# UNCERTAINTY-AWARE 3D RECONSTRUCTION FOR DYNAMIC UNDERWATER SCENES

**Rui Liu**[*]    **Zhibo Duan**[*]    **Jianzhe Gao**    **Wenguan Wang**[†]    **Yi Yang**

The State Key Lab of Brain-Machine Intelligence, Zhejiang University, Hangzhou, China
https://underwater-dynamic-field.github.io/UDF

## ABSTRACT

Underwater 3D reconstruction remains challenging due to the intricate interplay between light scattering and environment dynamics. While existing methods yield plausible reconstruction with rigid scene assumptions, they struggle to capture temporal dynamics and remain sensitive to observation noise. In this work, we propose an Uncertainty-aware Dynamic Field (UDF) that jointly represents underwater structure and view-dependent medium over time. A canonical underwater representation is initialized using a set of 3D Gaussians embedded in a volumetric medium field. Then we map this representation into a 4D neural voxel space and encode spatial-temporal features by querying the voxels. Based on these features, a deformation network and a medium offset network are proposed to model transformations of Gaussians and time-conditioned updates to medium properties, respectively. To address input-dependent noise, we model per-pixel uncertainty guided by surface-view radiance ambiguity and inter-frame scene flow inconsistency. This uncertainty is incorporated into the rendering loss to suppress the noise from low-confidence observations during training. Experiments on both controlled and in-the-wild underwater datasets demonstrate our method achieves both high-quality reconstruction and novel view synthesis.

## 1 INTRODUCTION

**Background.** Underwater 3D reconstruction is a foundational yet underexplored problem across diverse applications, including marine ecological monitoring [1] and autonomous navigation [2–4]. Recent advances in 3D reconstruction, such as neural radiance fields (NeRF) [5] and 3D Gaussian Splatting (3DGS) [6], have demonstrated remarkable ability to recover geometry and synthesize photorealistic views across both static and dynamic scenes. However, these methods are typically developed for clear-air environments, assuming a zero-density medium [7]. In underwater imaging, both object surfaces and the participating water volume affect image intensities [8]. As a result, directly applying vanilla NeRF or 3DGS to underwater data confuses geometric cues with medium-induced attenuation, leading to degraded reconstruction quality.

**Motivation.** Underwater imaging involves a complex interplay between *light scattering* and *environment dynamics* [9]. Specifically, the radiance along a camera ray comprises direct reflection from scene surfaces and accumulated scattering from surrounding water volume. Recent studies on underwater reconstruction either adopt learnable medium parameters during rendering [10, 11], or employ binary motion masks for moving objects [12–14]. While these methods yield promising results, two key challenges remain (Fig. 1). ❶ Underwater environments often exhibit complex dynamic phenomena, such as object motion, surface deformations, and time-varying medium properties. As the participating medium induces wavelength-dependent attenuation and scattering, modeling scene geometry and appearance under such non-rigid dynamic scenes is non-trivial [15]. ❷ Real-world underwater observations, especially from in-the-wild videos, suffer from degraded visibility and random perturbations [14, 16, 17]. Due to varying noise levels across viewpoints and frames, recovering geometry from such uncontrolled data will introduce ghosting artifacts and temporal in-

---

[*]The first two authors contribute equally to this work.
[†]Corresponding author: Wenguan Wang.

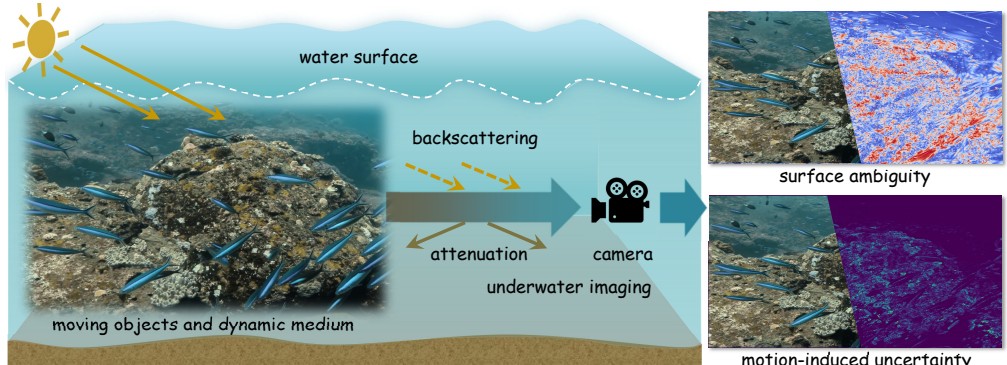

Figure 1: Challenges in dynamic underwater reconstruction. Underwater scenes contain moving objects and dynamic participating medium, where wavelength-dependent attenuation and backscattering jointly degrade visibility. Our UDF explicitly models surface ambiguity and motion-induced uncertainty, which highlight regions where observations are unreliable. The rays are schematic and do not depict exact physical light paths (see §1).

consistency [18–20]. However, most reconstruction methods [10–12, 14] in underwater environments overlook this input-dependent uncertainty, limiting their adaptability to ambiguous regions.

**Method.** To address these challenges, we propose an Uncertainty-aware Dynamic Field (UDF) for underwater reconstruction, built upon three key designs: *i*) a *unified dynamic underwater field* that jointly represents time-varying 3D geometry and the participating medium, *ii*) *motion-aware medium dynamics* conditioned on scene motion, and *iii*) an *underwater-specific heteroscedastic uncertainty* formulation that explicitly models input-dependent ambiguity. For ❶, our unified dynamic underwater field captures the evolution of both Gaussian-based structure and view-dependent medium properties. Specifically, a canonical underwater representation is initialized using a set of 3D Gaussian ellipsoids embedded within a volumetric medium field (§3.1). Each Gaussian serves as an explicit geometric primitive, while the surrounding water volume is modeled by a spectral neural field conditioned on ray directions. To encode spatial-temporal features of underwater dynamics, we map this representation into a compact 4D neural voxel space via planar factorization [21]. Built upon this representation, a deformation network is proposed to predict both spatial motion and shape transformation of the 3D Gaussians. Meanwhile, the time-varying volumetric medium properties are updated based on motion vectors via a medium offset network (§3.2). This design enables a consistent representation of dynamic geometry and motion-aware medium for underwater reconstruction.

For ❷, we incorporate the heteroscedastic uncertainty into the rendering loss to mitigate the impact of input-dependent noise (§3.3). To estimate per-pixel confidence, we analyze two physically grounded cues, *i.e.*, surface-view radiance ambiguity and inter-frame scene flow inconsistency. The first factor captures radiance ambiguity when the ray direction is closely aligned with the surface normal. In this case, scattering contributions are hard to disentangle from direct radiance, and high-frequency appearance variations can be misinterpreted as geometry [22–24]. To quantify this view-dependent factor, we estimate pixel-wise radiance confidence using the angle discrepancy between the ray direction and a pseudo-normal derived from the depth gradient. The second factor accounts for temporal instability caused by inter-frame inconsistencies. Appearance shifts or non-uniform motion across views result in ambiguous correspondence along motion trajectories. We estimate this factor using a Horn-Schunck flow-guided [25] approximation of scene dynamics. These two cues are combined into a per-pixel variance term within a probabilistic rendering loss. This allows the model to adaptively suppress the noise in ambiguous regions while maintaining physical plausibility.

**Results.** Our method is evaluated on both controlled underwater datasets [10] and in-the-wild video sequences [12, 26] (§4). UDF achieves high-fidelity reconstruction across diverse scenes, *e.g.*, **30.17** PSNR and **0.885** SSIM on DRUVA[26], and **27.72** PSNR and **0.883** SSIM on NUSR[12]. Furthermore, UDF enables high-quality novel view synthesis under adverse visibility conditions.

## 2    RELATED WORK

**NeRF and 3DGS for Static/Dynamic Scenes.** NeRF [5] and its extensions [27–29] have demonstrated remarkable performance in static scene reconstruction and generation [30–36], but suffers

from high rendering cost and slow optimization. Later, 3DGS [6, 37] is proposed as an explicit alternative, enabling real-time rendering via rasterization of Gaussian primitives. While early NeRF and 3DGS focus on static scenes [38], recent efforts extend them to dynamic scenarios [39–41]. Dynamic NeRF approaches either encode time as input [42, 43] or decouple scene motion via deformation fields [44, 45]. 4D Gaussian Splatting (4DGS) methods [20, 46–58] model temporal dynamics by either deforming static Gaussian primitives or directly learning continuous spatial-temporal fields. Some works are further proposed to enhance these frameworks, including dynamic surface modeling [59–62], temporal embeddings [63], and tracking-based correspondence [64, 65]. Despite these advances, most existing approaches operate under the clear-air assumption and fail to explicitly model medium-induced effects, thereby limiting their generalization to real-world underwater environments.

**Underwater Scene Reconstruction.** Underwater environments present unique challenges for 3D reconstruction due to light attenuation and scattering [66]. Traditional approaches address these degradations via model-free color correction [67–69] or physics-based enhancement [8, 16, 70, 71] on single images without explicit scene geometry modeling. Existing methods [10, 72–74] incorporate underwater light propagation into the volume rendering process. They adopt the revised underwater image formation model [9] to separate direct radiance from backscattering components within the NeRF pipeline. To improve reconstruction efficiency, some studies [11, 75–77] combine 3D Gaussian primitives with learnable medium features in underwater settings. Beyond static modeling, recent advances adopt dynamic NeRF [12, 13] or motion-aware 3DGS with binary masks [14] to handle dynamic elements. UDR-GS [78] directly employs 4DGS [46] with depth priors for basic motion modeling, yet ignores the influence of the participating medium during rendering. In this paper, we propose a unified dynamic field that jointly models underwater structure and medium over time, while incorporating uncertainty-aware optimization for underwater reconstruction.

**Uncertainty in Reconstruction.** Quantifying uncertainty is essential to improve the robustness of 3D scene reconstruction [18, 79–81]. For NeRF-based methods, early efforts adopt Bayesian formulations to estimate per-ray predictive variance [18, 82–85], providing probabilistic interpretations of scene geometry. Recent methods attempt to integrate uncertainty into 3DGS-based frameworks, including variational inference [86–88], Fisher information [89], deep feature-based estimation [19], and uncertainty-aware regularization [20, 90]. However, few methods address uncertainty under dynamic conditions, where both geometry and appearance evolve over time. In contrast, our method incorporates per-pixel uncertainty modeling by combining surface-view radiance ambiguity and inter-frame flow inconsistency. These cues are integrated into a heteroscedastic rendering loss, allowing the model to suppress the noise from low-confidence observations during training.

## 3 METHOD

**Overview.** Given a set of RGB images with camera poses and normalized timestamps, our goal is to reconstruct dynamic underwater scenes by jointly modeling scene geometry, medium effects, and their temporal evolution. UDF contains 3D Gaussians $\mathcal{G}$, a neural medium module $\mathcal{F}^{\mathrm{m}}$, spatial-temporal encoding, a deformation network $\mathcal{D}^{\mathrm{g}}$, and a medium offset network $\mathcal{D}^{\mathrm{m}}$ (Fig. 2). A canonical underwater representation is initialized using a set of 3D Gaussian ellipsoids $\mathcal{G}$ embedded within a volumetric medium field $\mathcal{F}^{\mathrm{m}}$ (§3.1). To model temporal dynamics, we encode this representation into a spatial-temporal space. The time-conditioned networks $\mathcal{D}^{\mathrm{g}}$ and $\mathcal{D}^{\mathrm{m}}$ are used to predict the evolution of Gaussians and medium attributes, respectively (§3.2). To address observation noise, we incorporate heteroscedastic uncertainty into the rendering loss based on surface-view radiance ambiguity and inter-frame flow inconsistency (§3.3). Implementation details are provided in §3.4.

### 3.1 CANONICAL UNDERWATER REPRESENTATION

**Volume Rendering with Medium.** To model the underwater scenes, the physically realistic medium term is incorporated into the volume rendering equation [10, 91, 92]. Given an image pixel $\boldsymbol{x}^{\mathrm{2d}} \in \mathbb{R}^2$, the corresponding camera ray is defined as $\boldsymbol{r}(s) = \boldsymbol{o} + s\boldsymbol{\omega}$, where $\boldsymbol{o}$ denotes the camera center and $\boldsymbol{\omega}$ is the ray direction. The color radiance $C(\boldsymbol{r})$ along this ray results from the accumulated contributions of both underwater structure surfaces and surrounding water medium [10, 91, 92]:

$$C(\boldsymbol{r}) = C^{\mathrm{str}}(\boldsymbol{r}) + C^{\mathrm{med}}(\boldsymbol{r}) = \int_0^\infty T(s)\big(\sigma^{\mathrm{str}}(s)c^{\mathrm{str}}(s) + \sigma^{\mathrm{med}}(s)c^{\mathrm{med}}(s)\big)ds, \tag{1}$$

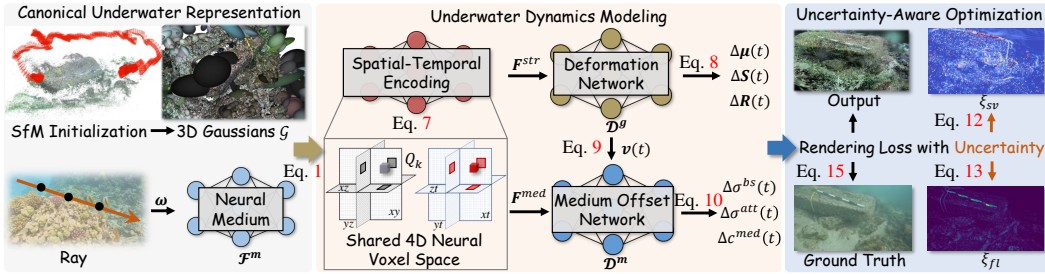

Figure 2: Overview of UDF (§3). UDF begins with a canonical underwater representation composed of 3D Gaussian primitives $\mathcal{G}$ initialized from SfM, and a neural medium $\mathcal{F}^m$ conditioned on the ray direction $\boldsymbol{\omega}$ (§3.1). This representation is then encoded into a shared spatial-temporal space as $\boldsymbol{F}^{\mathrm{str}}$ and $\boldsymbol{F}^{med}$. To model scene dynamics, a deformation network $\mathcal{D}^g$ and a medium offset network $\mathcal{D}^m$ are introduced to predict the evolution of both Gaussians and medium attributes (§3.2). To mitigate observation noise, an uncertainty-aware rendering loss is incorporated based on surface-view radiance ambiguity $\xi_{sv}$ and inter-frame flow inconsistency $\xi_{fl}$ (§3.3).

where $\sigma^{\mathrm{str}}(s)$ and $c^{\mathrm{str}}(s)$ represent the density and emitted color of scene structure at distance $s$, while $\sigma^{\mathrm{med}}(s)$ and $c^{\mathrm{med}}(s)$ are for the volumetric medium. The transmittance term $T(s)$ encodes the attenuation of light due to the participating medium:

$$T(s) = \exp\left(-\int_0^s \left(\sigma^{\mathrm{str}}(s') + \sigma^{\mathrm{med}}(s')\right) ds'\right) = T^{\mathrm{str}}(s) \cdot T^{\mathrm{med}}(s). \tag{2}$$

This formulation captures the interplay between geometry-induced occlusion and medium-induced scattering. In the following, we parameterize $\sigma^{\mathrm{str}}(s)$ and $c^{\mathrm{str}}(s)$ using Gaussian primitives, and model $\sigma^{\mathrm{med}}(s)$ and $c^{\mathrm{med}}(s)$ via a learnable neural field (see Eq. 4 & 5).

**Underwater Representation Initialization.** Given the input underwater images, we first calibrate camera poses and recover a sparse 3D point cloud using structure-from-motion (SfM) [93]. Based on this point cloud, a set of 3D Gaussian ellipsoids $\mathcal{G}$ is initialized to represent the canonical scene geometry. This SfM-based initialization is sufficient for our method (see Appendix §A.1), while recent work [20] provides additional Gaussians for dynamic regions. Each Gaussian primitive is based on a 3D Gaussian kernel $G(\boldsymbol{x})$ with a central position $\boldsymbol{\mu} \in \mathbb{R}^3$ and covariance matrix $\boldsymbol{\Sigma} \in \mathbb{R}^{3\times3}$:

$$G(\boldsymbol{x}^{\mathrm{3d}}) = \exp\left(-\frac{1}{2}(\boldsymbol{x}^{\mathrm{3d}} - \boldsymbol{\mu})^\top \boldsymbol{\Sigma}^{-1}(\boldsymbol{x}^{\mathrm{3d}} - \boldsymbol{\mu})\right), \quad \boldsymbol{\Sigma} = \boldsymbol{R}\boldsymbol{S}\boldsymbol{S}^\top\boldsymbol{R}^\top, \tag{3}$$

where $\boldsymbol{x}^{\mathrm{3d}} \in \mathbb{R}^3$ denotes a 3D location. $\boldsymbol{\Sigma}$ is decomposed into two learnable components, *i.e.*, a rotation matrix $\boldsymbol{R} \in \mathbb{R}^{3\times3}$ and a diagonal scaling matrix $\boldsymbol{S} \in \mathbb{R}^{3\times3}$. For the pixel $\boldsymbol{x}^{\mathrm{2d}}$ associated with ray $\boldsymbol{r}$, $N$ depth-ordered points $\{s_n\}_{n=1}^N$ along the ray intersect with the pixel using $\alpha$-blending:

$$C^{\mathrm{str}}(\boldsymbol{r}) = \sum_n T_n(s)\alpha_n c_n, \quad \text{where } \alpha_n = \sigma_n \exp\left(-\frac{1}{2}(\boldsymbol{x}^{\mathrm{2d}} - \boldsymbol{\mu}_n^{\mathrm{2d}})^\top (\boldsymbol{\Sigma}^{\mathrm{2d}})^{-1}(\boldsymbol{x}^{\mathrm{2d}} - \boldsymbol{\mu}_n^{\mathrm{2d}})\right), \tag{4}$$

where $\alpha_n$ denotes the opacity, and the color $c_n$ is modeled via spherical harmonics. The 3D Gaussian ellipsoids discretize the ray $\boldsymbol{r}(s)$ into a set of intervals, *i.e.*, $\{[0, s_1], \cdots, [s_{N-1}, s_N], [s_N, \infty]\}$, where $[s_N, \infty]$ accounts for background contributions beyond the last Gaussian.

For the volumetric medium, we introduce a neural medium model $\mathcal{F}^{\mathrm{m}}$ conditioned on spherical harmonic [23] encoded viewing directions $\boldsymbol{\omega}$ for medium properties $c^{\mathrm{med}}$ and $\sigma^{\mathrm{med}}$ along each ray $\boldsymbol{r}$ (see Eq. 10 for dynamic medium properties). To model wavelength-dependent underwater effects, $\sigma^{\mathrm{med}}$ is a three-channel density vector corresponding to RGB channels. The accumulated contribution from the volumetric medium $C^{\mathrm{med}}(\boldsymbol{r})$ is calculated by integrating within each ray segment [11]:

$$C^{\mathrm{med}}(\boldsymbol{r}) = \sum_n \int_{s_{n-1}}^{s_n} T_n(s)\sigma^{\mathrm{med}}c^{\mathrm{med}}ds. \tag{5}$$

In Eq. 4&5, the transmittance $T_n(s)$ is factorized into two terms [11], *i.e.*, accumulated occlusion from previous Gaussians $T^{\mathrm{str}}(s)$ and the exponential attenuation from the volumetric medium $T^{\mathrm{med}}(s)$:

$$T_n(s) = T^{\mathrm{str}}(s) \cdot T^{\mathrm{med}}(s) = \prod_{j=1}^{n-1}(1 - \alpha_j) \cdot \exp(-\sigma^{\mathrm{med}}s). \tag{6}$$

For compactness, the above expression uses a single medium coefficient $\sigma^{\mathrm{med}}$. In our actual formulation, this term is explicitly decomposed into an attenuation coefficient $\boldsymbol{\sigma}^{\mathrm{att}}$ and a backscatter coefficient $\boldsymbol{\sigma}^{\mathrm{bs}}$ following underwater imaging models [9, 10]. Both $\boldsymbol{\sigma}^{\mathrm{att}}$ and $\boldsymbol{\sigma}^{\mathrm{bs}}$ are parameterized as RGB-channel vectors, so the transmittance and emission are computed per color channel.

## 3.2 DYNAMIC FIELD FOR UNDERWATER SCENES

**Underwater Spatial-Temporal Features.** To model temporal changes in both underwater structures and surrounding medium effects, rather than assuming a static scene [11, 12], we extend the canonical underwater representation into a shared 4D neural voxel space via planar factorization [21]. This 4D space serves as a shared spatio–temporal backbone that jointly represents dynamic geometry and the time-varying participating medium. Given a 3D spatial coordinate $\boldsymbol{x}^{3d} = (x, y, z)$ and a normalized timestamp $t \in [0, 1]$, we decompose the 4D space-time domain into six orthogonal 2D planes $\{(x, y), (x, z), (y, z), (x, t), (y, t), (z, t)\}$. For both the Gaussian-based scene structure and the neural medium field in underwater representation space, each plane is respectively associated with learnable feature maps $\boldsymbol{f}_k^{str}$ and $\boldsymbol{f}_k^{med}$. Given a spatio-temporal query $Q(\boldsymbol{x}^{3d}, t)$, bilinear interpolation $\psi_{bl}$ is performed on each plane to retrieve local features:

$$\boldsymbol{F}^{str}(\boldsymbol{x}^{3d}, t) = \prod_k \psi_{bl}\big(\boldsymbol{f}_k^{str}, Q_k(\boldsymbol{x}^{3d}, t)\big), \quad \boldsymbol{F}^{med}(\boldsymbol{x}^{3d}, t) = \prod_k \psi_{bl}\big(\boldsymbol{f}_k^{med}, Q_k(\boldsymbol{x}^{3d}, t)\big), \tag{7}$$

where $Q_k(\boldsymbol{x}^{3d}, t)$ denotes the 4D query point of the $k$-th plane. These interpolated features from different planes are fused by element-wise multiplication. $\boldsymbol{F}^{str}(\boldsymbol{x}, t)$ and $\boldsymbol{F}^{med}(\boldsymbol{x}, t)$ encode spatio-temporal features of geometry and medium properties for dynamic motion and appearance modeling.

**Joint Geometry and Medium Dynamics.** To capture the dynamics of scene geometry and motion-aware medium, we introduce a time-conditioned deformation module built upon the spatio-temporal features $\boldsymbol{F}^{str}$ and $\boldsymbol{F}^{med}$. Different from vanilla 4DGS [46], this module estimates both geometric transformations of the 3D Gaussian primitives and the updates of the medium attributes. For each canonical Gaussian centered at $\boldsymbol{\mu}$, we employ a deformation network $\mathcal{D}^g$ to predict temporal offsets in translation $\Delta\boldsymbol{\mu}(t) \in \mathbb{R}^3$, scaling $\Delta\boldsymbol{S}(t) \in \mathbb{R}^3$, and rotation $\Delta\boldsymbol{R}(t) \in \mathbb{R}^3$, respectively:

$$\Delta\boldsymbol{\mu}(t), \Delta\boldsymbol{S}(t), \Delta\boldsymbol{R}(t) = \mathcal{D}^g(\boldsymbol{F}^{str}(\boldsymbol{\mu}, t), \boldsymbol{F}^{med}(\boldsymbol{\mu}, t)). \tag{8}$$

These parameters define the deformation of each Gaussian from the canonical space to its time-dependent state. By conditioning on both structural and medium-aware features, the deformation network is able to model plausible underwater dynamics. For the evolving state of the participating medium, a medium offset network $\mathcal{D}^m$ is introduced to estimate dynamic properties. A 2D scene flow vector $\boldsymbol{v}(t)$ is calculated by projecting temporal difference of Gaussian centers on the image plane:

$$\boldsymbol{v}(t) = \frac{\text{Proj}(\boldsymbol{\mu} + \Delta\boldsymbol{\mu}(t + \Delta t)) - \text{Proj}(\boldsymbol{\mu} + \Delta\boldsymbol{\mu}(t))}{\Delta t}, \tag{9}$$

where $\text{Proj}(\cdot)$ denotes the projection function from the 3D world space to the 2D image plane. $\boldsymbol{v}(t)$ encodes the motion cues of the underwater structure (represented by 3D Gaussians) on the image plane. The medium offset network $\mathcal{D}^m$ yields time-varying medium properties at location $x$, where two MLPs are used to integrate the motion cues based on $\boldsymbol{v}(t)$ and $\boldsymbol{\omega}$, respectively:

$$c^{med}(t), \sigma^{att}(t), \sigma^{bs}(t) = \mathcal{D}^m(\boldsymbol{F}^{med}(\boldsymbol{x}, t), \boldsymbol{v}(t), \boldsymbol{\omega}). \tag{10}$$

This enables complete modeling of environment dynamics, thereby improving reconstruction accuracy.

## 3.3 UNCERTAINTY-AWARE OPTIMIZATION

In underwater environments, degraded visibility and random perturbations often introduce noise into observations. Such noise is inherently input-dependent and varies with surface geometry and scene motion, leading to inconsistent geometry and appearance. To address this, we incorporate an uncertainty-aware rendering loss that adaptively modulates per-pixel supervision based on surface-view radiance ambiguity and inter-frame flow inconsistency (see visualization in Appendix §A.1).

**Surface-View Radiance Ambiguity.** This factor captures radiance ambiguity when the ray direction is closely aligned with the surface normal. In such cases, scattering contributions are hard to disentangle from direct radiance, and high-frequency appearance variations are misinterpreted as geometry cues [22, 23, 94–96]. To approximate this ambiguity, we compute a pseudo-normal vector $\boldsymbol{n}$ based on the gradient of the corresponding depth map $\boldsymbol{D}$ [97]. Each pixel of $\boldsymbol{D}$ is calculated by the expected distance $d(\boldsymbol{r})$ using $\alpha$-blending (similar to Eq. 4):

$$\boldsymbol{n} \approx \frac{[-\frac{\partial \boldsymbol{D}}{\partial x}, -\frac{\partial \boldsymbol{D}}{\partial y}, 1]}{\sqrt{(\frac{\partial \boldsymbol{D}}{\partial x})^2 + (\frac{\partial \boldsymbol{D}}{\partial y})^2 + 1}}, \quad d(\boldsymbol{r}) = \frac{\sum_{n=1}^N T_n(t)\alpha_n d_n}{\sum_{n=1}^N T_n(t)\alpha_n}. \tag{11}$$

We define the surface-view uncertainty based on the pseudo-normal $\boldsymbol{n}$ and the ray direction $\boldsymbol{\omega}$:

$$\xi_{\text{sv}}^2 = (\max(0, \boldsymbol{\omega} \cdot \boldsymbol{n}))^2. \tag{12}$$

$\xi_{\text{sv}}^2$ is used only as a soft cue in the variance term of Eq. 15: it modulates the weight of the photometric loss, but does not directly change the rendered color or geometry. Furthermore, the ablation study in Table 5 indicates the overall robustness of UDF does not rely on perfect pseudo-normal estimates.

**Inter-Frame Flow Inconsistency.** Non-rigid scene or appearance shifts often lead to inter-frame inconsistencies, resulting in artifacts during reconstruction. To measure such inconsistencies, we estimate a per-pixel motion vector $\boldsymbol{v}_{\text{pixel}}$ by aggregating the scene flow of visible Gaussians in Eq. 9, and evaluate its alignment with image gradients inspired by the Horn-Schunck optical flow [25]:

$$\xi_{\text{fl}}^2 = \left(\nabla I \cdot \boldsymbol{v}_{\text{pixel}} + \frac{\partial I}{\partial t}\right)^2 + \epsilon_0, \quad \boldsymbol{v}_{\text{pixel}} = \frac{\sum_{n=1}^{N} T_n(t)\alpha_n \boldsymbol{v}_n}{\sum_{n=1}^{N} T_n(t)\alpha_n}, \tag{13}$$

where $\nabla I$ denotes the spatial image gradient, $\frac{\partial I}{\partial t}$ is the temporal derivative of intensity, $\epsilon_0$ is a small constant for numerical stability, and $\boldsymbol{v}_n$ is the motion vector of the $n$-th Gaussian.

**Probabilistic Rendering Loss.** We formulate the rendering objective as probabilistic regression, where each pixel color is modeled as a normal distribution [18, 19]. The total uncertainty $\xi_{\text{total}}^2$ combines both $\xi_{\text{sv}}^2$ and $\xi_{\text{fl}}^2$. The rendering loss $\mathcal{L}_{\text{c}}$ is defined as the negative log-likelihood of the ground truth color $\hat{C}(\boldsymbol{r})$ under a predicted normal distribution with mean $C(\boldsymbol{r})$ and variance $\xi_{\text{total}}^2$:

$$\mathcal{L}_{\text{c}} = \frac{\|\hat{C}(\boldsymbol{r}) - C(\boldsymbol{r})\|^2}{2\xi_{\text{total}}^2} + \frac{1}{2}\log\xi_{\text{total}}^2, \quad \text{where} \ \ \xi_{\text{total}}^2 = \xi_{\text{sv}}^2 + \xi_{\text{fl}}^2. \tag{14}$$

This probabilistic formulation enables the model to down-weight ambiguous observations, instead of applying a uniform deterministic per-pixel loss to all measurements as in prior methods [11, 12, 78], leading to more stable and temporally consistent reconstruction over time. A total variation term $\mathcal{L}_{\text{tv}}$ is applied to the total loss $\mathcal{L}$ for smoothness of the predicted color sequence [46, 61]:

$$\mathcal{L} = \mathcal{L}_{\text{c}} + \mathcal{L}_{\text{tv}}. \tag{15}$$

### 3.4 Implementation Details

**Network Architecture.** UDF contains 3D Gaussians $\mathcal{G}$, a neural medium module $\mathcal{F}^{\text{m}}$, spatial-temporal encoding, a deformation network $\mathcal{D}^{\text{g}}$, and a medium offset network $\mathcal{D}^{\text{m}}$ (§3). The neural medium model $\mathcal{F}^{\text{m}}$ adopts spherical harmonic encoding and employ an MLP with 128 hidden units and Sigmoid activation to predict per-ray medium properties. For spatial-temporal encoding, we use planar factorization and learnable 4D voxels implemented via K-planes [21]. For the deformation network $\mathcal{D}^{\text{g}}$ (Eq. 8) and the medium offset network $\mathcal{D}^{\text{m}}$ (Eq. 10), we use separate MLPs to compute the temporal evolution of both 3DGS geometry and medium parameters.

**Training and Inference.** We first optimize Gaussian primitives $\mathcal{G}$ and the spectral medium field $\mathcal{F}^{\text{m}}$ for warm-up. Then we jointly train the deformation network $\mathcal{D}^{\text{g}}$ and the medium offset module $\mathcal{D}^{\text{m}}$ with the uncertainty-aware rendering loss (§3.3). We set $\epsilon_0 = 1 \times 10^{-4}$ for numerical stability (Eq. 13). To reduce the overhead of Horn-Schunck flow estimation, we approximate the second term of Eq. 13 using simplified inter-frame difference. We use the Adam optimizer with an initial learning rate of $1 \times 10^{-3}$ which decays exponentially to $1.5 \times 10^{-4}$. During inference, uncertainty modeling is disabled and the rendering module is used to produce the final output.

**Reproducibility.** Experiments are conducted with an Intel i9-14900K and an NVIDIA RTX 4090 GPU. 3DGS rasterisation is implemented in CUDA, while the remaining pipeline is developed using PyTorch. We use the gradient clip to stabilize training.

## 4 Experiment

**Datasets.** Our method is evaluated on both controlled and in-the-wild underwater datasets:

- **NUSR** [12] is collected from in-the-wild captured clips from the Internet, containing four monocular underwater videos with dynamic objects: Turtle, Coral, Composite, and Sardine.

- **DRUVA** [26] includes video sequences of 20 different artifacts. All videos are captured under natural illumination at a depth of $3 - 6$ m from the sea surface with a resolution of $1920 \times 1080$.
- **SeaThru** [10] contains real-world scenes from IUI3-Red Sea, Japanese Gardens, Panama, and Curaçao including 29, 20, 18, and 20 images respectively, each with a resolution of $900 \times 1400$.

**Evaluation Metrics.** Following existing methods [11, 12], we primarily assess our experimental results using peak-signal-to-noise ratio (PSNR), perceptual quality measure LPIPS [98], and structural similarity index (SSIM) [99]. All reported results are averaged over five runs on the same test set.

**Baselines.** We compare our UDF with recent approaches. For NUSR [12], we adopt static reconstruction methods, such as Instant-NGP [100], SeaThru-NeRF [10], as well as dynamic methods, including DynamicNeRF [13], NUSR [12], 4DGS [46], and WaterSplatting [11]. For DRUVA [26], the baseline methods include SeaThru-NeRF [10], TiNeuVox [101], 4DGS [46, 52], and WaterSplatting [11]. For SeaThru [10], we evaluate 3DGS[6], SeaThru-NeRF [10], NUSR [12], WaterSplatting [11], and 4DGS [46]. All methods are reproduced using their publicly available implementations.

## 4.1 QUANTITATIVE RESULTS

**Performance on NUSR.** Table 1 reports quantitative results of UDF on NUSR [12]. Compared to the original NUSR model [12], UDF achieves consistently better reconstruction quality. For instance, in the *Turtle* scene, it improves PSNR from 28.10 to **33.73**, increases SSIM from 0.899 to **0.965**, and significantly reduces LPIPS from 0.216 to **0.051**, demonstrating the effectiveness of our uncertainty-guided reconstruction strategy. Similar improvements are also observed in other scenes.

Table 1: Quantitative results on NUSR [12] (§4.1).

| Models | Sardine | | | Coral | | | Turtle | | | Composite | | |
|---|---|---|---|---|---|---|---|---|---|---|---|---|
| | PSNR↑ | SSIM↑ | LPIPS↓ | PSNR↑ | SSIM↑ | LPIPS↓ | PSNR↑ | SSIM↑ | LPIPS↓ | PSNR↑ | SSIM↑ | LPIPS↓ |
| Instant-NGP [100] [SIGGRAPH 2022] | 21.73 | 0.850 | 0.467 | 20.87 | 0.438 | 0.730 | 26.42 | 0.874 | 0.225 | 22.81 | 0.596 | 0.571 |
| DynamicNeRF [13] [ICCV2021] | 19.70 | 0.676 | 0.690 | 17.77 | 0.542 | 0.826 | 23.31 | 0.837 | 0.426 | 16.27 | 0.738 | 0.476 |
| SeaThru-NeRF [10] [CVPR 2023] | 21.37 | 0.578 | 0.606 | 23.89 | 0.648 | 0.405 | 27.06 | 0.881 | 0.191 | 16.21 | 0.405 | 0.828 |
| 4DGS [46] [CVPR 2024] | 21.42 | 0.627 | 0.486 | 25.81 | 0.769 | 0.258 | 25.06 | 0.838 | 0.239 | 26.81 | 0.844 | 0.201 |
| NUSR [12] [CVPR 2024] | 21.58 | 0.720 | 0.454 | 26.17 | 0.828 | 0.157 | 28.10 | 0.899 | 0.216 | 25.09 | 0.799 | 0.238 |
| WaterSplatting [11] [3DV 2025] | 23.16 | **0.876** | 0.184 | 28.67 | **0.908** | 0.098 | 25.20 | 0.879 | 0.233 | 27.20 | 0.910 | 0.124 |
| Ours | **24.41** | 0.847 | **0.168** | **28.72** | 0.890 | **0.085** | **33.73** | **0.965** | **0.051** | **27.53** | **0.911** | **0.090** |
| Error Bars (±) | (0.02) | (<1e-3) | (<1e-3) | (0.02) | (<1e-3) | (<1e-3) | (0.05) | (<1e-3) | (<1e-3) | (0.01) | (<1e-3) | (<1e-3) |

**Performance on DRUVA.** Table 2 presents results on representative scenes from the DRUVA dataset. Compared to WaterSplatting [11], UDF consistently achieves superior performance across PSNR, SSIM, and LPIPS. Notably, in A11, UDF improves PSNR from 32.33 to **34.03** and SSIM from 0.962 to **0.964**, and reduces LPIPS from 0.207 to **0.123**, verifying the effectiveness of dynamic modeling. Similar improvements across other scenes further underscore the general applicability of UDF.

Table 2: Quantitative results on DRUVA [26] (§4.1).

| Models | A1 | | | A2 | | | A11 | | | A12 | | |
|---|---|---|---|---|---|---|---|---|---|---|---|---|
| | PSNR↑ | SSIM↑ | LPIPS↓ | PSNR↑ | SSIM↑ | LPIPS↓ | PSNR↑ | SSIM↑ | LPIPS↓ | PSNR↑ | SSIM↑ | LPIPS↓ |
| TiNeuVox [101] [SIGGRAPH 2022] | 23.05 | 0.757 | 0.664 | 21.74 | **0.843** | 0.637 | 27.02 | 0.878 | 0.355 | 27.59 | 0.911 | 0.221 |
| SeaThru-NeRF [10] [CVPR 2023] | 26.27 | 0.778 | 0.336 | 20.23 | 0.694 | 0.481 | 27.62 | 0.916 | 0.254 | 26.13 | 0.883 | 0.282 |
| 4DGaussian [52] [ICLR 2024] | 22.84 | 0.688 | 0.535 | 20.93 | 0.586 | 0.638 | 26.84 | 0.882 | 0.273 | 25.84 | 0.906 | 0.244 |
| 4DGS [46] [CVPR 2024] | 20.75 | 0.736 | 0.449 | 19.54 | 0.678 | 0.516 | 24.09 | 0.938 | 0.173 | 25.79 | 0.956 | 0.128 |
| WaterSplatting [11] [3DV 2025] | 28.71 | 0.830 | 0.376 | 26.33 | 0.809 | 0.406 | 32.33 | 0.962 | 0.207 | 30.86 | 0.958 | 0.201 |
| Ours | **29.29** | 0.842 | **0.300** | **27.35** | 0.801 | **0.384** | **34.03** | **0.964** | **0.123** | **31.01** | **0.969** | **0.100** |
| Error Bars (±) | (0.12) | (<1e-3) | (0.001) | (0.06) | (<1e-3) | (<1e-3) | (0.11) | (<1e-3) | (<1e-3) | (0.09) | (<1e-3) | (<1e-3) |

**Performance on SeaThru.** Table 3 reports quantitative results of UDF on the SeaThru dataset. Compared to 4DGS [46], our method achieves substantial improvements across all metrics. For instance, in the *Panama* scene, UDF improves PSNR from 25.90 to **32.95**, increases SSIM from 0.801 to **0.930**, and reduces LPIPS from 0.277 to **0.065**, which highlights the effectiveness of incorporating the MLP-encoded medium representation in improving underwater reconstruction quality. On average, our method achieves consistently higher PSNR and lower LPIPS than all baselines, while SSIM is slightly lower than WaterSplatting [11] on a few specific sequences. As SSIM primarily measures local structural similarity, the sequences where WaterSplatting [11] attains marginally higher SSIM scores typically contain static content. Therefore, this static-style method can already preserve most

local background structures. In such cases, the advantages of UDF in dynamic modeling are not fully reflected in SSIM when the overall structural layout can be well reconstructed by a static method.

Table 3: Quantitative results on SeaThru [10] (§4.1).

| Models | CURAÇAO | | | IUI3-RedSea | | | Panama | | | Japanese Gardens | | |
|---|---|---|---|---|---|---|---|---|---|---|---|---|
| | PSNR↑ | SSIM↑ | LPIPS↓ | PSNR↑ | SSIM↑ | LPIPS↓ | PSNR↑ | SSIM↑ | LPIPS↓ | PSNR↑ | SSIM↑ | LPIPS↓ |
| 3DGaussians [6] [TOG 2023] | 28.31 | 0.873 | 0.221 | 22.98 | 0.843 | 0.245 | 29.20 | 0.893 | 0.152 | 21.49 | 0.854 | 0.216 |
| SeaThru-NeRF [10] [CVPR 2023] | 30.19 | 0.873 | 0.210 | 25.90 | 0.785 | 0.304 | 27.84 | 0.834 | 0.224 | 21.84 | 0.767 | 0.249 |
| NUSR [12] [CVPR 2024] | 30.03 | 0.822 | 0.238 | 22.70 | 0.624 | 0.347 | 23.75 | 0.686 | 0.263 | 25.81 | 0.853 | 0.182 |
| 4DGS [46] [CVPR 2024] | 24.75 | 0.706 | 0.195 | 22.54 | 0.630 | 0.473 | 25.90 | 0.801 | 0.277 | 22.42 | 0.779 | 0.323 |
| WaterSplatting [11] [3DV 2025] | 32.20 | **0.948** | 0.116 | 29.84 | **0.889** | 0.203 | 31.61 | **0.942** | 0.080 | 24.74 | 0.892 | 0.116 |
| Ours | **33.82** | 0.940 | **0.071** | **31.09** | 0.830 | **0.139** | **32.95** | 0.930 | **0.065** | **27.39** | **0.899** | **0.073** |
| Error Bars (±) | (0.03) | (<1e-3) | (<1e-3) | (0.05) | (<1e-3) | (<1e-3) | (0.01) | (<1e-3) | (<1e-3) | (0.04) | (<1e-3) | (<1e-3) |

## 4.2 QUALITATIVE RESULTS

**Rendering Quality.** In Fig.3, we compare our method against SeaThru-NeRF [10] and WaterSplatting [11] on DRUVA [26]. In A11 scene, the yellow-boxed region highlights a calibration board. Our reconstruction closely resembles the ground-truth reference, preserving sharp edges and color fidelity. In A12 scene, the red-boxed area reveals a moving rope, and our method demonstrates better performance in reconstructing such dynamic structure (see more results in Appendix §A.1).

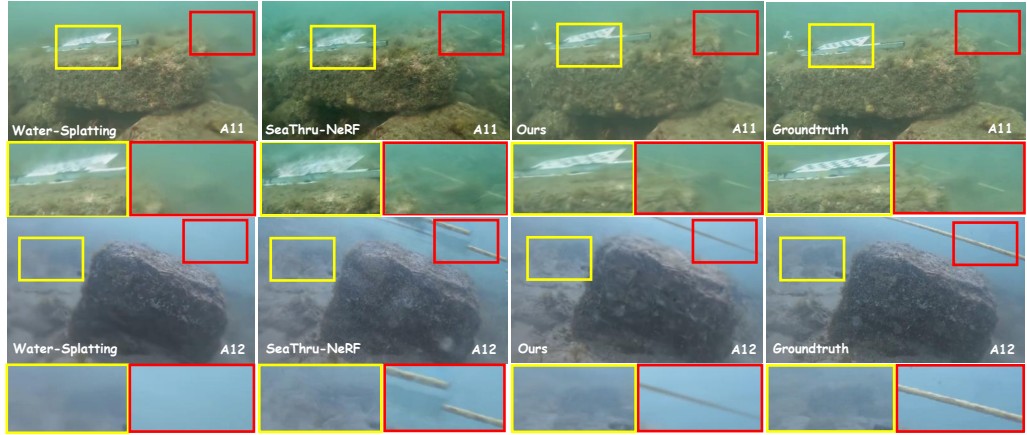

Figure 3: Underwater scene rendering on DRUVA [26]. We compare our method with SeaThru-NeRF [10] and WaterSplatting [11] on A11 and A12 scenes. For each method, we present four rendered images and provide two zoomed-in views of representative regions below each image, focusing on dynamic content and challenging areas. Our method produces consistently higher-quality results in both dynamic and static regions. (§4.2).

**Medium Removal and Depth Maps.** In Fig. 4, we provide medium-free renderings by disabling the neural medium module $\mathcal{F}^m$, and full rendered results with zoomed-in views. To further present geometry accuracy, we include the rendered depth maps (Eq. 11), which exhibit structural consistency with the scene layout. In *Turtle*, our approach accurately recovers the natural appearance of muddy terrain, producing visually realistic outputs. In contrast, other methods [11, 12] generate distorted colors after "medium removal", which amplifies small estimation errors and pushes colors toward unrealistic hues. By explicitly separating dynamic geometry and medium in a unified 4D field, and by using heteroscedastic uncertainty to down-weight ill-posed regions, UDF avoids such severe hue shifts and produces more plausible appearances (see more visualizations in Appendix §A.1).

**User Study on Novel View Synthesis.** To further validate the perceptual quality of novel view synthesis, we conduct a blind human evaluation study comparing the outputs of different methods. Concretely, 40 college students are invited to evaluate 1,500 rendered images. In each session, participants are randomly presented a triplet of renderings generated by SeaThru-NeRF [10], Water-Splatting [11], and our UDF, respectively. Each triplet contains multiple novel views of the same scene along with a ground-truth reference. Across all sessions, our method was selected as the top

choice in more than $90\%$ of cases. This indicates that UDF not only improves perceptual realism of underwater reconstructions but also maintains consistent visual quality across diverse viewpoints.

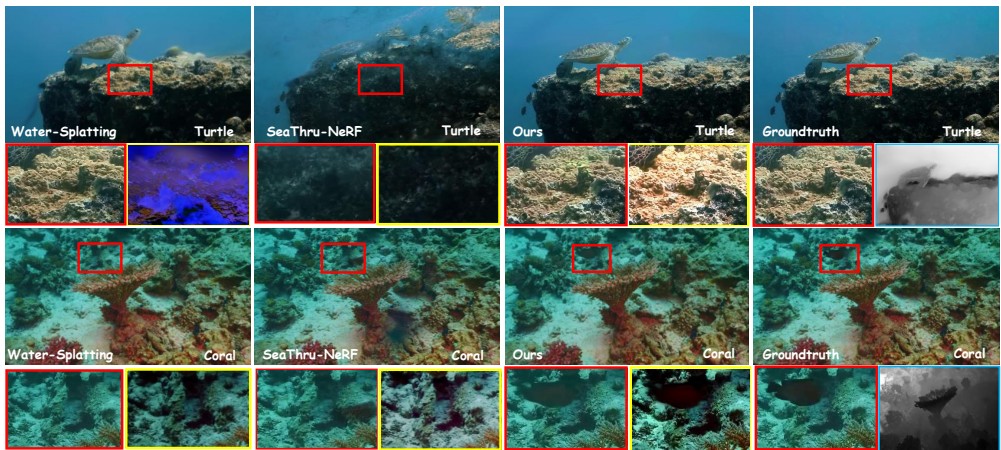

Figure 4: Scene Rendering and Medium Removal on NUSR [12]. We compare our method with SeaThru-NeRF [10] and WaterSplatting [11]. Zoomed-in regions show the original rendering results. Leveraging our composite underwater representation, we can remove medium effects by disabling the medium module (yellow boxes), recovering colors for both dynamic and static objects. We also include the rendered depth maps in the bottom-right corner of the groundtruth, where brighter values indicate greater distances (§4.2).

## 4.3 DIAGNOSTIC EXPERIMENT

**Model Design.** Table 4 presents an ablation study on the neural medium module ($\mathcal{F}^m$), the offset network ($\mathcal{D}^m$), and the deformation network ($\mathcal{D}^g$) on NUSR [12]. Removing $\mathcal{D}^m$ entirely leads to a notable performance drop, with PSNR falling to 30.84, SSIM decreasing to 0.951, and LPIPS rising to 0.072. Excluding $\mathcal{F}^m$ results in moderate degradation, indicating that modeling temporal variation in medium properties contributes to reconstruction fidelity. In addition, removing $\mathcal{D}^g$ causes a considerable decline in performance, with PSNR reduced from 33.73 to 31.46, highlighting the necessity of modeling dynamic geometry for accurate reconstruction. These results validate the importance of all three components in improving both photometric accuracy and temporal coherence across frames.

Table 4: Ablation study of model design (§4.3).

| $\mathcal{F}^m$ | $\mathcal{D}^g$ | $\mathcal{D}^m$ | NUSR Turtle [12] | | |
|---|---|---|---|---|---|
| | | | PSNR↑ | SSIM↑ | LPIPS↓ |
| − | − | − | 29.90 | 0.948 | 0.075 |
| ✓ | − | ✓ | 31.46 | 0.953 | 0.068 |
| − | ✓ | ✓ | 31.53 | 0.954 | 0.052 |
| ✓ | ✓ | − | 30.84 | 0.951 | 0.072 |
| ✓ | ✓ | ✓ | **33.73** | **0.965** | **0.051** |

**Uncertainty-Aware Optimization.** Table 5 presents an ablation study on the proposed uncertainty modeling components (Eq. 14). Removing the entire uncertainty terms results in significant performance degradation, with PSNR dropping from 33.73 to 30.83, SSIM from 0.965 to 0.941, and LPIPS increasing from 0.051 to 0.070. Excluding the surface-view radiance ambiguity term $\xi_{sv}^2$ leads to a moderate decline (PSNR: 32.48 and SSIM: 0.951), while omitting the inter-frame flow inconsistency term $\xi_{fl}^2$ yields comparable degradation (PSNR: 31.52 and SSIM: 0.954). These results demonstrate that both $\xi_{sv}^2$ and $\xi_{fl}^2$ are essential for improving image quality and ensuring consistent dynamics across frames.

Table 5: Ablation study of uncertainty (§4.3).

| $\xi_{sv}^2$ | $\xi_{fl}^2$ | NUSR Turtle [12] | | |
|---|---|---|---|---|
| | | PSNR↑ | SSIM↑ | LPIPS↓ |
| − | − | 30.83 | 0.941 | 0.070 |
| ✓ | − | 31.52 | 0.954 | 0.063 |
| − | ✓ | 32.48 | 0.951 | 0.062 |
| ✓ | ✓ | **33.73** | **0.965** | **0.051** |

**Parameter Sensitivity.** We conduct a sensitivity analysis to evaluate the impact of key hyperparameters across all test scenes. First, varying the hidden layer size of the neural medium module $\mathcal{F}^m$ among $\{64, 128, 256\}$ results in an average PSNR fluctuation of only 0.2 dB. Adjusting the weight of the uncertainty loss term (Eq. 14) within a range leads to PSNR variations of less than 0.5 dB. For 3D Gaussians $\mathcal{G}$, hyperparameters such as the total number of Gaussians and the splitting threshold are loosely constrained by scene complexity. For example, decreasing the splitting threshold from $1e-4$

to $5e-5$ yields consistent performance with negligible impact on reconstruction accuracy. These findings indicate that our model exhibits low sensitivity to hyperparameter settings, demonstrating stability and ease of deployment in diverse underwater reconstruction scenarios.

**Initialization Strategy.** In our experiments (§4.1), the traditional SfM method, *i.e.*, COLMAP [93], proves effective for 3DGS initialization. To further evaluate alternatives, we explore the recent VGGT [102] and observe comparable reconstruction quality in Table 6. This demonstrates that our pipeline remains robust across different initialization choices.

Table 6: Analysis on different initialization (§4.3).

| Models | NUSR [12] | | | SeaThru [10] | | |
|---|---|---|---|---|---|---|
| | PSNR↑ | SSIM↑ | LPIPS↓ | PSNR↑ | SSIM↑ | LPIPS↓ |
| VGGT [102] | 28.27 | **0.908** | 0.103 | **31.33** | **0.902** | 0.082 |
| COLMAP [93] | **28.59** | 0.903 | **0.098** | 31.31 | 0.899 | **0.087** |

**Robustness to Calibration Inaccuracies.** We additionally organize an ablation study that perturbs input camera poses with Gaussian noise in Table 7. We evaluate three noise levels (1%, 3%, 5%) and report the average performance under these perturbations. We observe that the degradation under noisy poses is small, and UDF still maintains strong reconstruction quality, which further supports its robustness to calibration inaccuracies in underwater capture.

Table 7: Robustness to Camera Calibration (§4.3).

| | DRUVA A2 [26] | | SeaThru [10] Panama | |
|---|---|---|---|---|
| | PSNR↑ | SSIM↑ | PSNR↑ | SSIM↑ |
| with Gaussian noise | 27.10 | 0.793 | 32.79 | 0.912 |
| without noise (ours) | **27.35** | **0.801** | **32.95** | **0.930** |

**Runtime and Memory Analysis.** We evaluate the runtime efficiency and memory usage of our method on DRUVA [26] in Table 8. Our method achieves **23.81** FPS while consuming 10.70 GB of GPU memory, with a parameter size of 33.67 MB. In comparison, SeaThru-NeRF [10] operates slower at only 0.43 FPS. WaterSplatting [11] consumes even more memory (12.93 GB) while operating at a lower frame rate (18.55 FPS). Although 4DGS [46] achieves the highest speed (29.27 FPS), it comes with a moderately larger parameter size (54.75 MB) and lower reconstruction metrics. In terms of training efficiency, our method converges in $18,000$ iterations. Overall, UDF provides a favorable trade-off between speed, memory usage, and model size, making it suitable for real-time or resource-constrained deployment.

Table 8: Efficiency analysis on DRUVA [26] (§4.3).

| Models | Param. Size (MB) | GPU Mem. (GB) | FPS |
|---|---|---|---|
| SeaThru-NeRF [10] | 201.32 | 11.70 | 0.43 |
| WaterSplatting [11] | 21.49 | 12.93 | 18.55 |
| 4DGS [46] | 54.75 | 2.88 | 29.27 |
| Ours | 33.67 | 10.70 | 23.81 |

**Failure Cases.** In Fig. 5, we present a failure case in NUSR [12]. A representative region is cropped and enlarged for detailed inspection. The yellow and magenta boxes highlight two failure modes: blurry reconstruction in distant regions and artifacts in fine structures. These issues are primarily caused by severe depth-dependent light attenuation, which is inherent to underwater environments. Such attenuation significantly reduces the visibility of distant scene content, increasing the difficulty of accurately modeling both geometry and appearance (see more failure cases in Appendix A.1).

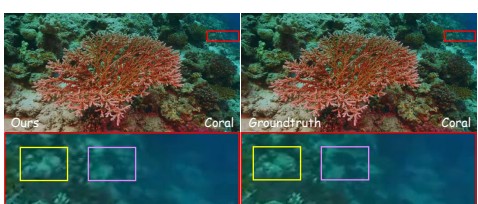

Figure 5: Failure cases in NUSR [12] (§4.2).

## 5 CONCLUSION

We presented an uncertainty-aware dynamic field (UDF) for underwater 3D reconstruction, addressing the inherent challenges posed by light scattering and environmental dynamics. Our method jointly models dynamic geometry and medium properties using 4D encoding, and introduces per-pixel heteroscedastic uncertainty to adaptively mitigate observation noise. We quantify input-dependent uncertainty by leveraging surface-view radiance ambiguity and inter-frame flow inconsistency, and further incorporate it into the rendering loss. Experimental results demonstrate that UDF achieves high-fidelity reconstruction and novel view synthesis under adverse visibility conditions.

**Acknowledgment.** This work was supported by Zhejiang Provincial Natural Science Foundation of China (No. LR26F020002), Fundamental Research Funds for the Central Universities (226-2025-00057), and CIE-Tencent Robotics XRhino-Bird Focused Research Program.

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

## A  APPENDIX

We first present additional visualizations (§A.1), including Uncertainty and Depth Maps (Fig. 6), Medium Decoupling and Removal (Fig. 7), More Failure Cases (Fig. 8), Dynamic Motion Trajectories (Fig. 9), and Point Clouds from SfM (Fig. 10). Furthermore, a detailed discussion (§A.2) is provided with Terms of use, Limitations, Future Direction, and Broader Impacts. Finally, we present additional comparisons (Table 9) and supplementary videos (§A.5).

### A.1  MORE VISUALIZATIONS AND ANALYSES

**Uncertainty and Depth Maps.** In Fig. 6, we visualize the depth map $D$, surface-view ambiguity $\xi^2_{sv}$, inter-frame flow inconsistency $\xi^2_{fl}$, and the corresponding input images. As shown on the right, the moving objects within the orange bounding box is clearly highlighted in the inter-frame flow inconsistency map, demonstrating the model's ability to capture dynamic motion cues. These areas typically exhibit high inconsistency due to non-rigid motion and temporal misalignment. Our method effectively distinguishes both geometry-induced and motion-induced uncertainty.

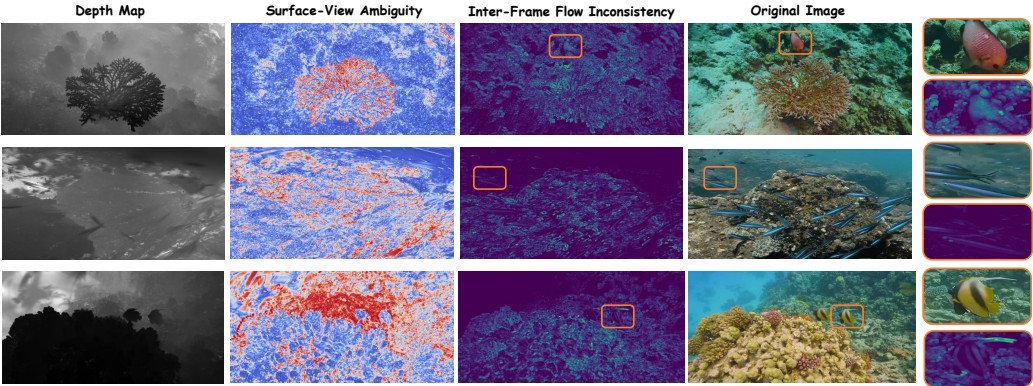

Figure 6: Uncertainty visualization on 'Coral', 'Sardine', and 'Composite' scenes of NUSR [12]. We show per-pixel uncertainty magnitude in our model using surface-view ambiguity and inter-frame flow inconsistency. Furthermore, the highlighted orange boxes outline dynamic regions in both the inter-frame flow inconsistency maps and the corresponding original images, which are then magnified and shown on the right.

**Medium Decoupling and Removal.** In Fig. 7, we visualize the decomposition of underwater rendering outputs for representative views in 'Coral' and 'Japanese Garden Red Sea' scenes. For each selected view, we display the estimated medium appearance, object appearance, restored appearance, the final rendered image, and the corresponding ground truth. Our model effectively disentangles medium-related effects, such as scattering and attenuation, from the underlying object surfaces. The restored appearance reveals faithful color recovery, while the estimated medium maps capture plausible distributions of underwater light propagation. These results demonstrate the effectiveness of our medium modeling in separating rendering components and enhancing reconstruction realism.

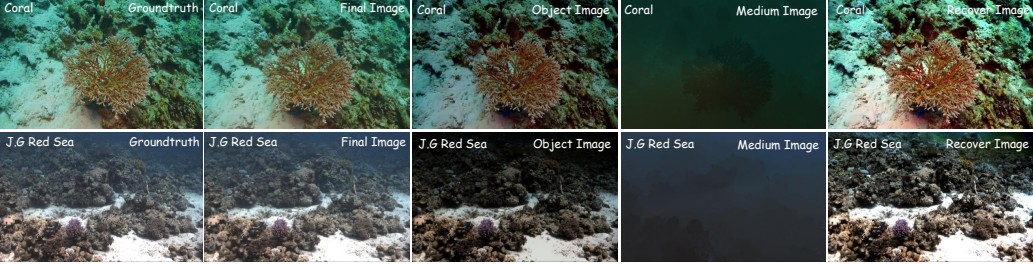

Figure 7: Medium removal visualization for underwater rendering in 'coral' and 'Japanese Garden Red Sea' scenes. We show rendering results from a selected viewpoint in each scene, including the estimated medium appearance, object appearance, restored appearance, final rendered image and ground truth. These visualizations demonstrate the effectiveness of our medium modeling and its contribution to accurate underwater reconstruction.

**More Failure Cases.** In Fig. 8, we present an additional failure case from NUSR[12]. While our method successfully captures the motion of the large foreground fish, it produces noticeable ghosting artifacts on a smaller red fish undergoing rapid movement (highlighted in the red box ). This failure reveals a limitation in handling fast-moving, small-scale objects with high-frequency motion.

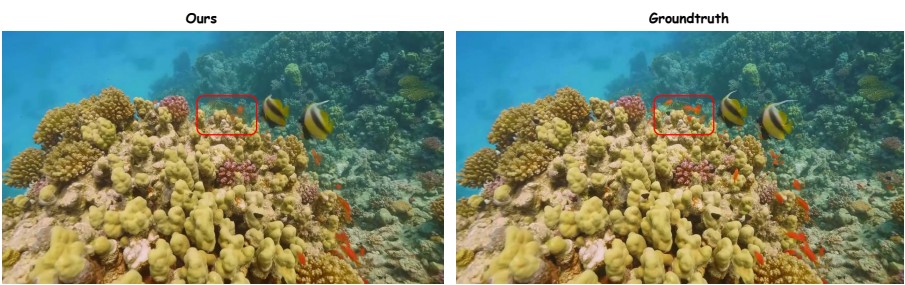

Figure 8: Failure cases in NUSR [12].

**Dynamic Motion Trajectories.** In Fig. 9, trajectories are densely distributed around dynamic objects such as the turtle, while static regions like rocks exhibit minimal motion. This spatial distribution aligns with intuitive expectations and highlight the necessity of modeling underwater dynamics.

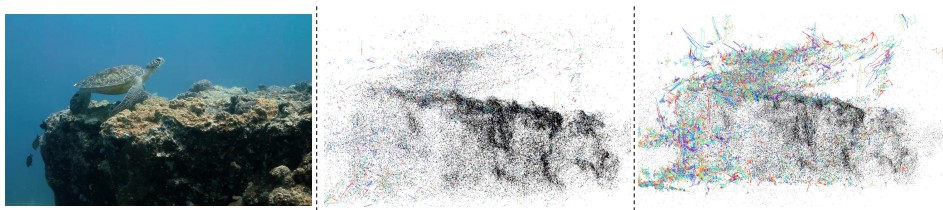

Figure 9: Motion trajectories in 'Turtle' scene of NUSR [12]. From left to right: input image, two-frame motion trajectories, and multi-frame motion trajectories.

**Point Clouds from SfM.** In Fig. 10, we present point clouds obtained from SfM [93] (§3.1). These point clouds exhibit high quality and faithfully capture the geometry of objects in the real scene, providing valuable priors for reliable Gaussian initialization.

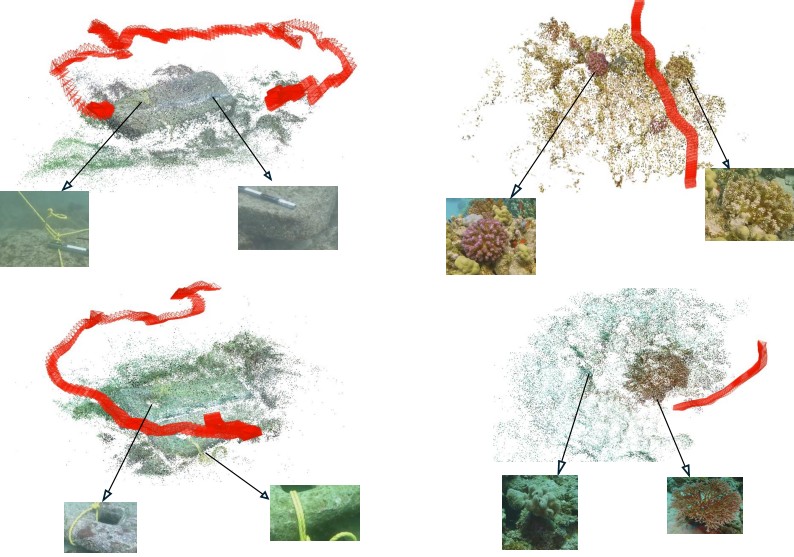

Figure 10: Point clouds for gaussian initialization.

**Assumption on Illumination Conditions.** Our method assumes that underwater images are captured under natural sunlight instead of active lighting systems. This assumption holds for all datasets used in our experiments, where ambient daylight provides relatively uniform illumination. This design simplifies modeling of wavelength-dependent attenuation and scattering, and works well for these datasets where natural light is the dominant source. We consider extending this to handle non-uniform, active lighting a promising direction for future work, potentially by inverse rendering with disentangled illumination representations [97, 103].

## A.2 DISCUSSION

**Terms of Use, Privacy, and License.** We do not collect or use any private or personally identifiable information in our study. All datasets used are anonymized and publicly available for research purposes. The algorithms described in this work are made available exclusively for academic research purposes. We conduct experiments using three publicly available underwater datasets: SeaThru [10], NUSR [12], and DRUVA [26]. The terms of use and licensing details for these datasets are summarized as follows. SeaThru[1] is released under the Apache License 2.0, which permits both commercial and non-commercial use, provided proper attribution and license preservation. NUSR provides data access through a public download link[2], but does not explicitly state a license. We use it solely for non-commercial academic research in compliance with common usage norms. DRUVA requires a signed release agreement and approval via a request form prior to access.[3] We obtained the dataset through the official process and comply with its academic-use conditions.

**Limitations.** While our method achieves strong performance in modeling dynamic underwater scenes with uncertainty-aware rendering, it has several limitations. (1) Real-world underwater environments, such as oceans, rivers, and lakes, exhibit diverse dynamics and medium properties. To fully assess the generalization of our method, more datasets covering a wider range of scenarios are needed. (2) Our method uses heteroscedastic aleatoric uncertainty to handle input-dependent noise but does not model epistemic uncertainty [79], which captures uncertainty in model parameters arising from limited data. Incorporating epistemic uncertainty could improve reliability in out-of-distribution settings. (3) Some bluish tint may still be observed in scenes, suggesting incomplete disentanglement between object appearance and medium effects. This residual tint arises primarily from relying exclusively on photometric supervision, which provides only pixel-level intensity information. The datasets used in our experiments are collected from in-the-wild Internet videos and lack auxiliary supervision, such as medium-free reference images or known transmission profiles. Without such prior knowledge, our model may struggle to fully decouple medium scattering effects, particularly in spectrally biased regions (*e.g.*, red-light attenuation in deeper water). This limitation highlights a valuable direction for future work: introducing auxiliary supervision via synthetic data with ground-truth albedo and transmission would provide explicit guidance, improving separation of object and medium. (4) As discussed in the failure cases (Appendix Fig. 8), our model can still produce artifacts around small, fast, highly deformable fish. These objects undergo rapid non-rigid motion at very fine spatial scales and are often observed from only a few viewpoints under strong scattering, making accurate reconstruction extremely challenging. Our primary goal is to jointly model dynamic geometry and dynamic underwater medium at the scene level. And extending the framework with fine-scale, object-aware motion modeling for such cases remains an important direction for future work.

**Future Direction.** A promising direction is to more explicitly address calibration inaccuracies that are typical in underwater capture (*e.g.*, refraction at the housing interface, mechanical tolerances, and slight pose drifts). A natural idea for future work is to jointly refine camera poses and refractive parameters within the dynamic field optimization, potentially coupled with uncertainty over camera extrinsics, to further improve robustness in severely miscalibrated settings. Beyond calibration, another important direction is the principled integration of physical priors into dynamic scene representations, enabling stronger generalization beyond the training data distribution. Incorporating multi-modal sensory signals—such as acoustic measurements or inertial priors—may further improve robustness in visually degraded or sensor-limited underwater settings. Moreover, uncertainty could be modeled more holistically, not only at the observation level but also within the representation and

---

[1]Dataset page: https://sea-thru-nerf.github.io

[2]NUSR download: https://drive.google.com/file/d/1AErUPPKwQ1cOwTF0ORVd6J91Od_Brw8z/view

[3]Dataset access via: https://github.com/nishavarghese15/DRUVA

rendering stages. This could be achieved through Bayesian modeling or variational inference [84], enabling the capture of epistemic uncertainty arising from model ambiguity. Furthermore, while the revised underwater image formation model [9] is designed for linear input data, our results, along with prior work [11, 12], suggest that neural approaches to medium modeling remain effective to both RGB images (*e.g.*, from NUSR [12] and DRUVA [26]) and linear data (*e.g.*, from SeaThru [8]). Further exploration is needed to generalize these findings across different datasets.

**Broader Impacts.** Our work enables reliable and physically grounded reconstruction in challenging underwater environments by jointly modeling geometric structures, medium dynamics, and observation uncertainty. This advancement holds significant potential for critical domains such as marine ecology, underwater archaeology, and autonomous robotics, where robust scene understanding is essential under degraded visual conditions. The deployment of such technologies in ecologically sensitive areas needs careful consideration. Ensuring transparency in data collection, fairness in system behavior, and adherence to ethical standards is crucial to prevent ecological disturbance or operational bias. We encourage the community to continue examining the environmental, safety, and ethical implications of exploration in underwater ecosystems such as the ocean.

## A.3 MORE RESULTS

Table 9: Quantitative results on SeaThru [10] (§A.3).

| Models | CURAÇAO | | | IUI3-RedSea | | | Panama | | | Japanese Gardens | | |
|---|---|---|---|---|---|---|---|---|---|---|---|---|
| | PSNR↑ | SSIM↑ | LPIPS↓ | PSNR↑ | SSIM↑ | LPIPS↓ | PSNR↑ | SSIM↑ | LPIPS↓ | PSNR↑ | SSIM↑ | LPIPS↓ |
| UW-GS [14] [WACV 2025] | 31.77 | **0.943** | 0.144 | 28.65 | **0.933** | 0.125 | 31.79 | 0.936 | 0.115 | 23.04 | 0.860 | 0.189 |
| SeaSplat [76] [Arxiv 2024] | 30.30 | 0.900 | 0.190 | 26.67 | 0.870 | 0.210 | 28.76 | 0.900 | 0.150 | 22.70 | 0.870 | 0.180 |
| Gaussian Splashing [77] [Arxiv 2024] | 31.26 | 0.920 | 0.170 | 24.73 | 0.920 | 0.110 | 31.35 | **0.940** | 0.110 | 24.73 | **0.920** | **0.110** |
| Ours | **33.82** | 0.940 | **0.071** | **31.09** | 0.830 | 0.139 | **32.95** | 0.930 | **0.065** | **27.39** | 0.899 | **0.073** |
| Error Bars (±) | (0.03) | (<1e-3) | (<1e-3) | (0.05) | (<1e-3) | (<1e-3) | (0.01) | (<1e-3) | (<1e-3) | (0.04) | (<1e-3) | (<1e-3) |

**Comparison on SeaThru.** In Table 9, we further compare against several recently proposed methods on SeaThru [10], including UW-GS [14], SeaSplat [76], and Gaussian Splashing [77]. Our method consistently outperforms these approaches across most metrics. For instance, in CURAÇAO, our method achieves a PSNR of **33.82**, significantly higher than UW-GS [14] (31.77), and obtains the lowest LPIPS score (**0.071**). Similar performance gains are observed across the other scenes, demonstrating the effectiveness and generalizability of our design.

## A.4 USE OF LARGE LANGUAGE MODELS

We did not use any large language models in this work.

## A.5 SUPPLEMENTARY MATERIAL

**Supplementary Videos.** We provide additional video visualizations in the supplementary material (filename: 'Supplementary Videos') to further demonstrate the effectiveness of our method. For each of three representative scenes, *i.e.*, IUI3-RedSea, Japanese Gardens from SeaThru [10], and Composite from NUSR [12], we render a sequence of images along the camera trajectories provided in the dataset and compile them into videos. Each video contains two side-by-side views: the left shows the final rendered results, while the right displays the estimated mediums. These visualizations highlight that our neural medium is temporally adaptive and effectively captures dynamic changes over time, contributing to improved scene reconstruction under diverse and challenging underwater conditions.

