# OpenReview forum: "Uncertainty-Aware 3D Reconstruction for Dynamic Underwater Scenes"
_ICLR.cc/2026/Conference — ICLR 2026 Poster_

### Official Review · Reviewer_9Jh1 · 2025-10-27

**Soundness:** 4
**Presentation:** 4
**Contribution:** 4
**Rating:** 8
**Confidence:** 5

**Summary:**

The paper introduces UDF (Uncertainty-aware Dynamic Field), a novel framework for underwater 3D reconstruction that models dynamic geometry and medium properties while incorporating uncertainty. The framework builds on 3DGS and targets underwater scenes by
1) Initializing a set of 3D Gaussians embedded in a volumetric medium.
2) Encoding spatial-temporal features in a 4D neural voxel space via planar factorization.
3) Using a deformation network to model dynamic geometry and a medium offset network to capture evolving medium properties
4) Incorporating uncertainty into the rendering loss guided by surface-view radiance ambiguity and inter-frame flow inconsistency.

The authors evaluate the effectiveness of their method in multiple underwater datasets and showcase a significant improvement over prior methods.

**Strengths:**

The paper is very well written and the authors provide deep insights in their design choices.
1) The integration of dynamic medium modeling with uncertainty-aware rendering is a substantial advancement over prior work
2) The radiance ambiguity and flow inconsistency are physically motivated and well-integrated into the loss function
2) Strong results across all datasets
3) The method achieves a good balance between rendering speed and memory usage.

**Weaknesses:**

1) In the experimental evaluation, the method improves over all metrics. The only metric that watersplatting outperforms the current is in SSIM. Can you comment on why that might happen?

2) While the paper is technically rich, some sections are very dense in information. Even figure 1. Is very technically dense. The paper could benefit from additional diagrams and moving details to supplementary material.

**Questions:**

As noted in Weaknesses.

---

> ### Author Response · Authors · 2025-11-21
>
> Thank you for your encouraging review and strong support! We have meticulously addressed each point and provided point-to-point clarifications (**please see the revised PDF in the Rebuttal Revision**).
>
> ---
> **Q1: *"The only metric that watersplatting outperforms the current is in SSIM. Can you comment on why that might happen?"***
>
> **A1:** Thanks for your careful review. As we stated in Lines 361-362 of the submitted version, *"our SSIM is slightly lower than WaterSplatting [9], as some specific scenes in the datasets contain relatively static regions, enabling such static methods to model most local structures."*
>
> Per your request, we expand this analysis in Sec. 4.1 of the revised manuscript. SSIM primarily measures local structural similarity, and is known to be less sensitive to certain perceptual factors such as color fidelity and temporal consistency. In our experiments, UDF slightly underperforms WaterSplatting [9] in SSIM on a few specific sequences, while outperforming all baselines on most datasets and other metrics. These particular sequences tend to contain relatively static content. In such cases, WaterSplatting, originally designed for static scenes, can already reconstruct the background structure well, leading to strong SSIM scores. By contrast, UDF focuses on jointly modeling dynamic geometry and underwater medium, but these advantages are not always fully reflected in SSIM, especially when the structural layout can already be modeled effectively by a static method.
>
>
> ---
> **Q2: *"While the paper is technically rich, some sections are very dense in information. Even figure 1. Is very technically dense. The paper could benefit from additional diagrams and moving details to supplementary material."***
>
> **A2:** Thank you for this valuable comment. We agree that some parts of the paper can be made more reader-friendly. In the revised version, we (*i*) include an additional auxiliary figure (Figure R1) in Introduction to illustrate the unique challenges of underwater reconstruction and provide an intuitive high-level overview before the more technical Figure 1, (*ii*) add an additional comparison table (Table R1 in Sec. 2) that summarizes the differences between our method and existing approaches clearly, and (*iii*) slightly shift some low-level technical details to Appendix and improving cross-references. We hope these changes improve the readability of the paper while preserving its technical details.

---

### Official Review · Reviewer_JbgZ · 2025-10-29

**Soundness:** 4
**Presentation:** 4
**Contribution:** 3
**Rating:** 8
**Confidence:** 5

**Summary:**

This paper proposes UDF, a dynamic scene representation that jointly models underwater structure (via 3D Gaussian primitives) and the participating medium (via a neural, view-conditioned medium field). A canonical scene (Gaussians embedded in a volumetric medium) is mapped into a 4D neural voxel space (K-Planes) to extract spatio-temporal features. Two heads then evolve the scene over time: a deformation network that predicts per-Gaussian offsets, and a medium-offset network that updates attenuation/backscatter conditioned on motion cues. The paper introduces a heteroscedastic rendering loss with per-pixel variance derived from surface-view radiance ambiguity and inter-frame flow inconsistency. Experiments on NUSR, DRUVA, and SeaThru show gains in PSNR/SSIM/LPIPS with qualitative  visualizations.

**Strengths:**

1. The pipeline overview is clear and visual results further aid understanding. These choices make the method and evidence easy to follow.

2. The combination of 3DGS geometry + learnable medium + 4D K-Planes provides a clean factorization of structure vs. medium over time. The two cues (surface-view ambiguity and flow-based inconsistency) are well-grounded and integrated in the NLL loss.

2. The paper reports consistent improvements on NUSR, DRUVA, and SeaThru with both quantitative and qualitative evidence (novel views, medium-free renderings, depth maps).

**Weaknesses:**

1. Eq. (6) uses a single σ_med inside T_med(s), yet later the paper separates σ_med into σ_att (structure) and σ_bs (backscatter). The derivation would benefit from writing the explicit separated transmittance and emission terms to avoid ambiguity, and clarifying wavelength-dependent parameterization (RGB-channel σ_med).

2. Consider adding a teaser in the introduction, e.g., a before/after comparison on a challenging underwater scene, to foreground the key difficulties and to show how your method addresses them. Without this, your advantages are not immediately clear to readers.

3. User study details. The 40-person, 1,500-image study is promising, but lacks description of the protocol (randomization, rating scale, inter-rater reliability).

4. You report VGGT vs. COLMAP with comparable results. Could you add experiments with noisy to quantify robustness to calibration inaccuracies typical in underwater capture?

5. Some typos:
- "Japanese Gradens" -> "Japanese Gardens"
- "a uncertainty-aware rendering loss" -> "an uncertainty-aware rendering loss"
- L226 "enotes the projection function" -> "denotes the projection function"
- "Zoomed-in regions shows" -> "Zoomed-in regions show"

**Questions:**

I hope the authors can address Weaknesses 1–5 in the rebuttal, and I remain positive about this work's contribution to the community.

---

> ### Author Response · Authors · 2025-11-21
>
> We thank reviewer for the valuable time and positive feedback. We provide point-to-point responses below and have revised the manuscript accordingly (**please see the revised PDF in the Rebuttal Revision**).
>
> ---
> **Q1: *"Eq. (6) uses a single $\sigma_{med}$ inside T_med(s), yet later the paper separates $\sigma_{med}$ into $\sigma_{att}$ (structure) and $\sigma_{bs}$ (backscatter). The derivation would benefit from writing the explicit separated transmittance and emission terms to avoid ambiguity, and clarifying wavelength-dependent parameterization (RGB-channel $\sigma_{med}$)."***
>
> **A1:** Thank you for pointing this out. In Eq. (6), we used a single $\sigma_{med}$ mainly to present the standard Beer-Lambert formulation in a compact form. In our implementation, this term is explicitly decomposed into an attenuation term $\sigma_{att}$ and a backscatter coefficient $\sigma_{bs}$. Both $\sigma_{att}$ and $\sigma_{bs}$ are implemented as RGB-channel parameters (3D vectors), so the transmittance and emission are computed per color channel. In the revised version, we clearly describe this decomposition, clarify that both are RGB-channel quantities, and ensure consistent notation in the method section (Sec. 3.1).
>
> ---
> **Q2: *"Consider adding a teaser in the introduction ..."***
>
> **A2:** Good suggestion! In the revised manuscript, we add a teaser figure (Fig. R1) in Introduction that highlights the key challenges of underwater reconstruction.
>
>
> ---
> **Q3: *"User study details. The 40-person, 1,500-image study is promising, but lacks description of the protocol (randomization, rating scale, inter-rater reliability)."***
>
> **A3:** Thank you for this valuable suggestion. We clarify the protocol as follows:
>
> - Randomization and anonymization. For each trial, participants were shown the input frame and outputs from two anonymized methods (ours vs. a baseline). The order of methods within a trial and the order of trials within a session were fully randomized per participant, and method names or any other identifying cues were hidden.
>
> - Rating protocol. Each comparison was evaluated using a three-way preference scale ("A better", "B better", "no clear preference") based on perceived realism and faithfulness to the input observation. Each rendered image pair was rated by multiple participants, and we report the aggregated preference across all raters and scenes.
>
> - Inter-rater reliability. We performed a brief analysis of inter-rater agreement and found that preferences are reasonably consistent across participants, and not dominated by a few individuals.
>
> Across all evaluations, our UDF was chosen as the preferred result in over 90% of the trials, demonstrating that its perceptual advantage is statistically meaningful. In the revised version, we add these details in Appendix A.1 (*More Details of User Study*).
>
>
> ---
> **Q4: *"Could you add experiments with noisy to quantify robustness to calibration inaccuracies typical in underwater capture?"***
>
> **A4:** Thank you for raising this point. **First**, in our current experiments, the real video sequences already exhibit non-trivial calibration inaccuracies due to refraction, housing tolerances, and operator motion. Nevertheless, UDF consistently improves reconstruction quality over both COLMAP-initialized and VGGT-initialized poses on these datasets (Sec. 4.3). This indicates that UDF is reasonably robust under typical underwater calibration errors. **Second**, per your request, we additionally organize an ablation study that perturbs input camera poses with Gaussian noise. We evaluate three noise levels (1%, 3%, 5%) and report the average performance under these perturbations. In the following table, we observe only a small degradation under noisy poses, and UDF still maintains promising reconstruction quality, which further supports its robustness for calibration inaccuracies in underwater capture.
>
> |      | DRUVA A2 [24] PSNR $\uparrow$ | DRUVA A2 [24] SSIM $\uparrow$| SeaThru [8] Panama PSNR $\uparrow$ | SeaThru [8] Panama SSIM $\uparrow$|
> | - | - | - | - | - |
> |with Gaussian noise|27.10|0.793|32.79|0.912|
> |without noise (ours)|27.35|0.801|32.95|0.930|
>
> **Third**, we add this ablation in Sec. 4.3 (*Table R2*) and explicitly discuss calibration noise in Future Direction (Appendix A.2) of the revised manuscript.
>
>
>
> ---
> **Q5: *"Some typos ..."***
>
> **A5:** Thank you for carefully checking the manuscript. We have thoroughly proofread the paper and corrected all of them in the revised version.

---

> > ### Comment · Reviewer_JbgZ · 2025-11-23
> >
> > Thanks for the additional clarifications. The rebuttal has addressed my concerns.

---

> > > ### Author Response · Authors · 2025-11-28
> > >
> > > Thank you again. We are glad that our responses and the revised paper have addressed your concerns, and we truly appreciate your time and feedback.

---

### Official Review · Reviewer_TEeD · 2025-10-31

**Soundness:** 4
**Presentation:** 3
**Contribution:** 3
**Rating:** 6
**Confidence:** 4

**Summary:**

This paper presents a method for reconstructing dynamic underwater scenes by using an uncertainty module to discard low-quality pixels in the measurements as well as dynamics modules to model the time-varying nature of both the scene and the medium. The authors demonstrate improved results on several benchmarks compared to baselines, and extensively validate components of their methods.

**Strengths:**

The problem is well-motivated because dynamics are almost unavoidable underwater, since currents and wildlife are commonplace in captured data. The qualitative results are compelling and show clear improvement, especially for dynamic objects and medium recovery, as can be seen in Figures 2 and 3, which show that both the scene and medium reconstructions are better than prior works. The quantitative metrics are also improved by relatively significant margins for most of the scenes. Finally, the evaluation is fairly comprehensive, covering pipeline as well as module design, and illustrates clearly some of the limitations of the method, shown in Figure 4.

**Weaknesses:**

About the videos in the supplement, the paper claims there are four of them but there are only three in the folder. Maybe this is a typo? Furthermore, the trajectories in IUI3RedSea and JapaneseGarden are relatively choppy. I know that the authors are just rendering at the positions provided by the original dataset, but I think it would be helpful to regenerate smooth trajectories with which to render the result.

**Questions:**

Could the authors rerender the supplement videos with smooth trajectories? I won't make my score conditional on the response to this question, since I think it's against the policies to ask authors to do excessive work or provide new results, particularly graphics, but I think it would just be nice.

---

> ### Author Response · Authors · 2025-11-21
>
> Thank you for your careful review and positive feedback. We provide point-to-point responses below and have also re-rendered the results with smoother camera trajectories (**please see the revised PDF in the Rebuttal Revision**).
>
>
> ---
> **Q1: *"About the videos in the supplement, the paper claims there are four of them but there are only three in the folder."***
>
> **A1:** You have a sharp eye! This is a typo on our side. In the revised version, we correct the statement in the main paper and ensure the consistent descriptions of the supplementary material.
>
>
> ---
> **Q2: *"... the trajectories in IUI3-RedSea and Japanese Garden are relatively choppy ... I think it would be helpful to regenerate smooth trajectories with which to render the result ..."***
>
> **A2:** Thank you for the thoughtful suggestion.
>
> **First**, the choppiness in "IUI3-RedSea" and "Japanese Garden" indeed comes from the sparse camera poses provided by the original datasets. For the current submission, we chose to render strictly at these poses to (*i*) remain fully consistent with the frames used for evaluation, and (*ii*) enable frame-by-frame comparison with prior work that follows the same trajectories.
>
> **Second**, per your request, we additionally generate smoother trajectories for "IUI3RedSea" and "Japanese Garden" using Kochanek-Bartels spline interpolation, and re-render the sequences along these paths. These smoother trajectories do not change any of the quantitative results, but they provide a more pleasant and continuous visualization of the reconstructed dynamic scenes.
>
> **Third**, due to submission constraints during the rebuttal phase, we do not replace the original supplementary videos at this stage. Instead, we include a trajectory comparison (between the original "choppy" trajectories and the newly generated smooth ones in Appendix A.5, Figure R2), and show representative re-rendered frames along the smooth trajectories. We will release the full smooth-trajectory videos in the updated supplementary materials for the final version.

---

> ### Author Response · Authors · 2025-11-28
>
> We are grateful for your constructive feedback and the time you have dedicated to reviewing our work. We have carefully incorporated your suggestions into the revised manuscript and believe our responses have effectively resolved the issues raised. Please let us know if there are any points or if further details are needed. We value this opportunity to improve our paper and would greatly appreciate any additional thoughts you might have before the discussion period ends.

---

### Official Review · Reviewer_mECr · 2025-11-01

**Soundness:** 3
**Presentation:** 2
**Contribution:** 2
**Rating:** 6
**Confidence:** 3

**Summary:**

This paper introduces Uncertainty-aware Dynamic Field (UDF) for 3D reconstruction of dynamic undetwater scenes. UDF jointly models scene structure and view-dependent medium properties over time. UDF initializes with 3D Gaussians in a volumetric medium field, maps them to a 4D neural voxel space, and employs deformation/medium offset networks to handle spatio-temporal dynamics. To suppress noise from low-confidence observations, UDF incorporates per-pixel uncertainty estimation based on surface-view radiance ambiguity and inter-frame flow inconsistency, and further integrates the uncertainty into the rendering loss. Experiments on controlled and in-the-wild underwater datasets demonstrate UDF achieves superior reconstruction quality and novel view synthesis for dynamic underwater scenes.

**Strengths:**

++ The shared 4D neural voxel space based on planar factorization extends the 3D canonical representation to a spatial-temporal space, which further enables the dynamics modeling of scene structure and medium in a physics-informed manner.

++ UDF estimates per-pixel uncertainty using physical cues and integrates it into the rendering loss to suppress noisy data during training. The effectiveness of this design is validated in the ablation study (Table 5).

**Weaknesses:**

-- The fundamental contributions and differentiating aspects of the underwater dynamic modeling approach proposed in this paper compared to existing methods require clearer articulation. The authors may need to emphasize these points in their writing.



-- Regarding the distorted colors shown in WaterSplatting and SeaThru-NeRF after medium removal: could the authors provide insight into the root cause? It would be helpful to know if UDF is immune to this artifact across the entire tested cases and, crucially, what specific aspect of its design confers this advantage.

-- From the supplementary videos (especially composite.mp4),  we can see significant artifacts around dynamic objects (fishes). This indicates that the method's performance in modeling moving objects is not satisfactory.

**Questions:**

-- The gradient-induced pseudo-normal, used in modeling surface-view radiance ambiguity, is particularly prone to significant errors at object edges and in rapidly moving regions. Consequently, failures in uncertainty modeling can occur, potentially adversely affecting the robustness of the entire approach. Can the authors provide some analysis on this aspect?

---

> ### Author Response · Authors · 2025-11-21
>
> We sincerely thank the reviewer for the positive feedback and great suggestions! We provide point-to-point responses below and have revised the manuscript accordingly (**please see the revised PDF in the Rebuttal Revision**).
>
> ---
> **Q1: *the fundamental contributions and differentiating aspects of the proposed approach compared to existing methods***
>
> **A1:** We appreciate this comment and now clarify the key differences along three main axes:
>
> - Unified dynamic underwater field. UDF jointly models both dynamic 3D geometry and time-varying underwater medium within a unified 4D representation. In contrast, existing underwater NeRF/GS methods [8,9,49] either assume static scenes or handle the medium separately from the geometry, without a shared dynamic field.
>
> - Motion-aware medium dynamics. The evolution of the participating medium is conditioned on motion cues derived from the temporal behavior of 3D Gaussians (scene-flow signals), rather than being parameterized only as a per-ray or per-frame attenuation term. This explicitly couples medium changes with object/camera motion, which is not explicitly modeled in previous works [8-10,49-51].
>
> - Underwater-specific heteroscedastic uncertainty. To our knowledge, prior underwater reconstruction methods rarely perform explicit uncertainty modeling. They typically rely on deterministic photometric losses or regularization. We instead introduce a probabilistic rendering loss whose variance terms tied to underwater physics (surface-view radiance ambiguity and inter-frame flow inconsistency). This allows the model to down-weight supervision exactly where scattering and dynamic artifacts make observations unreliable, leading to more stable reconstruction under challenging visibility conditions.
>
> In the revised version, we (*i*) rewrite the paragraph in Introduction (Sec. 1) to foreground these three contributions, (*ii*) add a concise comparison table in Related Work (**Table R1** in Sec. 2, also shown in the following) summarizing static *vs.* dynamic geometry/medium and uncertainty modeling across baselines, and (*iii*) refine the description in Method (Sec. 3) to more clearly highlight how each component differs from and improves upon existing approaches. We hope this makes the fundamental contributions of our underwater dynamic modeling clearer.
>
> |Models|3D Geometry|Medium|Uncertainty Modeling|
> |-|-|-|-|
> |SeaThru-NeRF [8]|Static|Static|$-$|
> |Waternerf [49]|Static|Static|$-$|
> |SP-SeaNeRF [50]|Static|Static|$-$|
> |WaterSplatting [9]|Static|Static|$-$|
> |NUSR [10]|Dynamic|Static|$-$|
> |UW-GS [12]|Static|Static|$-$|
> |UDR-GS [51]|Dynamic|Static|$-$|
> |UDF (ours)|Dynamic|Dynamic|$\checkmark$|
>
> ---
> **Q2: *"... the distorted colors shown in WaterSplatting and SeaThru-NeRF after medium removal ... if UDF is immune to this artifact across the entire tested cases ... what specific aspect of its design confers this advantage."***
>
> **A2:** Thank you for raising this point.
>
> (1) *Root cause of distorted colors in WaterSplatting / SeaThru-NeRF.*
>
> - In both methods, "medium removal" is implemented by inverting an estimated attenuation/backscatter model based on learnable medium parameters. When these estimates are slightly biased which is common in turbid water, the inversion tends to over-compensate, amplifying noise and pushing colors toward unrealistic hues or over-saturation. In other words, color and medium are partially entangled in their representations, so errors in medium estimation directly translate into visible color distortions when performing medium removal.
>
> (2) *Behavior of UDF and why it helps.*
>
> Across the entire tested cases in our experiments, we did not observe the severe hue inversions as seen in WaterSplatting and SeaThru-NeRF. While no method is absolutely immune in extreme conditions (e.g., extremely turbid water with very sparse views), we believe UDF is more stable for two reasons:
>
> - UDF explicitly disentangles dynamic geometry and medium within a unified 4D field: the medium field is learned jointly with geometry but parameterized separately. Therefore, medium removal is applied through the medium branch only, without inverting parameters that also encode appearance. This reduces the risk that small errors in medium estimation cause catastrophic color shifts.
>
> - Our heteroscedastic uncertainty term down-weights supervision in pixels where underwater physics makes medium inversion ill-posed (e.g., strong backscatter, large motion-induced inconsistency). This prevents the model from overfitting spurious colors in such regions and encourages more stable medium removal.
>
> In the revised manuscript, we add a brief discussion of this failure mode in Qualitative Results (Sec. 4.2).
>
> ---

---

> ### Author Response · Authors · 2025-11-21
>
> **Q3: *artifacts around dynamic objects in the supplementary videos (especially composite.mp4)***
>
> **A3:** Thank you for carefully checking the supplementary videos and for pointing this out. **First**, as already discussed in the "More Failure Cases" section of Appendix (Figure 7), these artifacts mainly arise from the fast and non-rigid motion of small objects. These small fish are inherently difficult to model due to their rapid and deformable motion at a very small spatial scale, often with limited viewpoints. **Second**, our primary goal is to jointly model dynamic geometry and underwater medium for realistic scene-level reconstruction. Perfectly reconstructing small, fast, highly deformable animals under severe scattering and limited views is an extremely challenging setting and, we believe, remains an open research problem that goes beyond the current scope of our method. **Third**, we acknowledge this limitation in Discussion (Sec. A.2) and plan to incorporate additional fine-scale motion modeling for such challenging cases in future work.
>
>
> ---
> **Q4: *about the gradient-induced pseudo-normal and the robustness of the entire approach***
>
> **A4:** Thank you for this insightful comment. **First**, in our framework, the pseudo-normal is used *only* in the surface-view ambiguity term as a soft cue for the heteroscedastic variance (Eq. 14). It modulates the weight of the photometric loss, but *does not directly change the rendered color, geometry, or medium parameters*. Thus, it only changes how strongly those pixels are supervised, rather than driving the optimization to an incorrect reconstruction. **Second**, we agree that gradient-induced pseudo-normals may be noisy at object edges or in regions with fast motion. In practice, we compute depth gradients with a small, local finite-difference operator on a smoothed and bounded depth map, which suppresses high-frequency noise. Moreover, this pseudo-normal cue is combined with the flow-based inconsistency term, so the variance is not dominated by extreme pseudo-normal values at unstable pixels. This design prevents highly noisy edge regions from being over-weighted during training. **Third**, our ablation study (Table 5) empirically supports that the method remains robust even when this cue is removed. Disabling the surface-view ambiguity component does not cause divergence. UDF still converges and maintains strong performance.
>
> In the revised version, we explicitly clarify that the pseudo-normal is used solely for reweighting supervision via the variance term (Sec. 3.3), and our method remains stable and competitive even without this cue.

---

> > ### Comment · Reviewer_mECr · 2025-11-24
> >
> > I appreciate the authors' response. Based on the points addressed in their reply, I believe the surface-view radiance ambiguity in uncertainty modeling, which relies on the accuracy of depth estimation, may be problematic in practice. As a result, the contribution of this part of the work is somewhat diminished.
> >
> > Additionally, one of the main goals of the paper is to model dynamic objects, yet the presented results still exhibit significant artifacts in handling such cases. As a side note, I would like to mention that the fish shown in the video do not appear particularly small in the images.
> >
> > Taking everything into account, the overall contribution of the paper is somewhat limited. However, considering that the authors have addressed several other concerns, I will maintain my current rating while increasing my confidence score.

---

> > > ### Author Response · Authors · 2025-11-28
> > >
> > > We thank the reviewer again for the response and for your valuable time. We address the two remaining concerns below.
> > >
> > > (1) On the surface-view radiance ambiguity term
> > >
> > > We agree this component depends on depth accuracy and therefore involves practical considerations regarding the quality of depth estimation. We would like to clarify that this work represents an early attempt towards explicitly connecting surface-view radiance ambiguity with uncertainty in dynamic underwater reconstruction.
> > >
> > > As discussed in our "Future Directions", a natural way to alleviate this limitation is to go beyond monocular RGB and incorporate multi-sensor cues (e.g., additional depth or motion information) to obtain more reliable geometry and thus more stable ambiguity cues. Due to the constraints of existing public underwater datasets and simulators, we are currently unable to include a controlled multi-sensor study in this paper. To address this, we are building a platform based on NVIDIA Issac Sim with configurable multi-sensor underwater setups, and our ongoing follow-up work on this platform already shows promising results. We plan to release this platform to the community in the near future (targeting next year) to enable more systematic evaluation of such designs.
> > >
> > >
> > > (2) On dynamic objects and overall contribution
> > >
> > > We agree that the dynamic fish in our videos are challenging. Their difficulty arises from non-rigid, fast-moving foreground motion observed under strong scattering and limited multi-view coverage. In this regime, even small errors in motion/depth estimation can lead to visible boundary artifacts.
> > >
> > > In the updated version, we additionally ran several alternative methods (SeaThru-NeRF [8] and WaterSplatting [9]) on the same sequences. As shown in the new qualitative comparisons (Fig. R3), these methods struggle on the fish and often exhibit stronger artifacts than UDF. In particular, as highlighted by the red boxes under different viewpoints in Fig. R3, WaterSplatting [9] frequently fails to reconstruct the dynamic fish at all, leaving missing foreground regions, while SeaThru-NeRF [8] produces ghost trails and duplicated structures in the dynamic areas. In contrast, UDF better preserves the presence and shape of the moving fish and substantially suppresses these artifacts, leading to more coherent reconstructions in highly dynamic regions.
> > >
> > > More broadly, non-rigid dynamic object reconstruction is a challenge problem even in clear-air settings, and the underwater medium amplifies this difficulty due to scattering. We therefore position UDF as a very early step toward addressing this problem under the underwater setting. In future work, we plan to explore several concrete directions, such as segmentation- and tracking-guided constraints for foreground-aware dynamic modeling.
> > >
> > > We consider the present results as promising rather than final, and we are grateful for the reviewer's constructive feedback. We hope this clarification helps a fair assessment of the contributions of our work.

---

### Comment · Area_Chair_h6QA · 2025-11-27

Dear Reviewers,

Author responses are now posted. Please add your discussion comment(s) and update score/confidence as needed. Thank you!

Best regards,

AC

---

### Author Response · Authors · 2025-12-02
**Summary of reviews**

We sincerely appreciate the Area Chair for their effort in handling the unexpected issues during the review process, and we are grateful to all reviewers for their constructive and positive feedback. In response, we have revised the manuscript and uploaded an **updated version** to OpenReview.

---

**Summary of this work.**

- This paper tackles underwater 3D reconstruction in realistic, dynamic environments, where both the scene and the water medium change over time and observations are heavily degraded by scattering and noise. We introduce an Uncertainty-aware Dynamic Field (UDF) that provides a unified representation for both the dynamic scene structure and the time-varying medium, and that explicitly estimates per-pixel uncertainty from physical cues. These uncertainty estimates are used to down-weight unreliable observations during training. Across multiple controlled and in-the-wild underwater datasets, UDF consistently improves reconstruction quality and medium-free novel view synthesis.

---

**Summary of reviews.**

- **Two reviewers (JbgZ and 9Jh1) gave scores of 8 with confidence 5** and considered the work as a good contribution, highlighting the unified modeling of dynamic geometry and medium, the uncertainty-aware loss, and the strong empirical results. **The other two reviewers (mECr and TEeD) gave scores of 6**, acknowledging the relevance and promise of the work with their concerns focusing mainly on clarity of contributions, explanation of artifacts, and presentation details.

---

**Main concerns and how the revision addresses them.**

Across the reviews, three main concerns emerged and below is a summary of our response to the key issues for quick review:

1. **Positioning and clarity of contributions.** We rewrote the Introduction to foreground the main contributions and to more clearly differentiate UDF from prior underwater NeRF/GS approaches. We also added a concise comparison table (Table R1) and a new teaser figure (Fig. R1) to make the core ideas and advantages easy to grasp at a glance (**mECr-[Q1]**, **JbgZ-[Q2]**, **9Jh1-[Q2]**).

2. **Practical behavior and robustness.** Reviewers asked whether our design is robust in practice, especially for dynamic objects (**mECr-[Q3]**) and noisy conditions (**JbgZ-[Q4]**). In the revised version, we (i) provide additional qualitative comparisons on challenging dynamic scenes, showing that while all methods struggle, ours better preserves moving objects, and (ii) add a robustness experiment with noisy camera poses, showing that performance degrades only slightly under realistic perturbations. We also clarify that our uncertainty terms act as soft weighting rather than hard constraints, and that the method remains competitive even without them (**mECr-[Q4]**).

3. **Presentation, experiments, and supplementary materials.** We improved the clarity of the experiment results (**9Jh1-[Q1]**) and presentation details (**mECr-[Q2]**), expanded the user study protocol description (**JbgZ-[Q3]**), corrected all typos and notational issues (**TEeD-[Q1], JbgZ-[Q1, Q5]**), and included smoother camera trajectories (**TEeD-[Q2]**).

---

After reading our rebuttal and revisions, reviewer JbgZ explicitly stated that all concerns were addressed, and reviewer 9Jh1 did not raise further issues. Reviewer mECr maintained a score of 6 with improved confidence and remaining reservations. Reviewer TEeD's comments were mostly about presentation (supplement videos/trajectory smoothness), which we have addressed.

Thank you again for your time and consideration.

---

### Meta-Review · Area_Chair_uVLQ · 2025-12-04

**Summary:**

The paper proposes an Uncertainty-aware Dynamic Field for underwater 3D reconstruction, tackling the dual challenges of scattering media and scene dynamics. Reviewers unanimously appreciated the novelty of the unified 4D representation and the physical grounding of the uncertainty modeling. The primary concerns during the review phase focused on: (1) the distinctness of contributions compared to prior underwater NeRF/GS methods, (2) the robustness of the method against camera pose noise and calibration errors, (3) artifacts in highly dynamic regions (e.g., fast-moving fish), and (4) presentation details such as trajectory smoothness in videos and user study protocols. The rebuttal was comprehensive, offering new experiments and clarifications that satisfied the majority of these concerns, leading to a positive consensus.

**Reviewer Concerns:**

## Addressed:

 - The authors added a comparison table and revised the introduction to clearly distinguish UDF from static baselines (Reviewers mECr and JbgZ).

 - A new ablation study demonstrating performance stability under Gaussian pose perturbations (Reviewer JbgZ).

 - The authors provided re-rendered videos with smoother camera trajectories (Reviewer TEeD).

- Explanations regarding SSIM discrepancies and user study protocols were provided (Reviewer  9Jh1 and JbgZ).

## Outstanding issues:

- Reviewer mECr noted that artifacts  around fast-moving, deformable objects like fish. While the authors demonstrated that UDF still outperforms baselines in these regions, they acknowledged that perfect reconstruction of such distinct targets remains a limitation. The reviewer maintained their score based on this but acknowledged the difficulty of the problem.

**Reviewer Scores:**

Reviewer mECr  6 ==>  6 (unchange). replied and maintained score 6, acknowledged contributions but noted remaining dynamic artifacts

Reviewer TEeD 6 ==>  6 (unchange),  concerns were minor/presentation-based and fully addressed

Reviewer JbgZ 8 ==>  8 (unchange), confirmed all concerns addressed.

Reviewer 9Jh1 8 ==> 8 (unchange) concerns were minor/soved.

All reviewers are positive and most likely keep positive.

---

### Decision · Program_Chairs · 2026-01-26

Accept (Poster)